https://doi.org/10.1038/s41467-021-26922-4　　**OPEN**

# *Chloranthus* genome provides insights into the early diversification of angiosperms

Xing Guo[1,11], Dongming Fang[1,11], Sunil Kumar Sahu [1,11], Shuai Yang[1], Xuanmin Guang[1], Ryan Folk [2], Stephen A. Smith [3], Andre S. Chanderbali[4], Sisi Chen[1,5], Min Liu[1], Ting Yang[1], Shouzhou Zhang [6], Xin Liu [1,7], Xun Xu [1,8], Pamela S. Soltis [4], Douglas E. Soltis [4,9✉] & Huan Liu [1,10✉]

Chloranthales remain the last major mesangiosperm lineage without a nuclear genome assembly. We therefore assemble a high-quality chromosome-level genome of *Chloranthus spicatus* to resolve enigmatic evolutionary relationships, as well as explore patterns of genome evolution among the major lineages of mesangiosperms (eudicots, monocots, magnoliids, Chloranthales, and Ceratophyllales). We find that synteny is highly conserved between genomic regions of *Amborella*, *Vitis*, and *Chloranthus*. We identify an ancient single whole-genome duplication (WGD) (κ) prior to the divergence of extant Chloranthales. Phylogenetic inference shows Chloranthales as sister to magnoliids. Furthermore, our analyses indicate that ancient hybridization may account for the incongruent phylogenetic placement of Chloranthales + magnoliids relative to monocots and eudicots in nuclear and chloroplast trees. Long genes and long introns are found to be prevalent in both Chloranthales and magnoliids compared to other angiosperms. Overall, our findings provide an improved context for understanding mesangiosperm relationships and evolution and contribute a valuable genomic resource for future investigations.

[1] State Key Laboratory of Agricultural Genomics, BGI-Shenzhen, Shenzhen 518083, China. [2] Department of Biological Sciences, Mississippi State University, Mississippi State, MS 39762, United States of America. [3] Department of Ecology and Evolutionary Biology, University of Michigan, Ann Arbor, MI 48103, United States of America. [4] Florida Museum of Natural History, University of Florida, Gainesville, FL, United States of America. [5] South China Botanical Garden, Chinese Academy of Sciences, Guangzhou, Guangdong 510650, China. [6] Laboratory of Southern Subtropical Plant Diversity, Fairy Lake Botanical Garden, Shenzhen, Chinese Academy of Sciences, Shenzhen 518004, China. [7] BGI-Fuyang, BGI-Shenzhen, Fuyang 236009, China. [8] Guangdong Provincial Key Laboratory of Genome Read and Write, BGI-Shenzhen, Shenzhen 518083, China. [9] Department of Biology, University of Florida, Gainesville, FL 32611, United States of America. [10] Department of Biology, University of Copenhagen, DK-2100 Copenhagen, Denmark. [11] These authors contributed equally: Xing Guo, Dongming Fang, Sunil Kumar Sahu. ✉email: dsoltis@ufl.edu; liuhuan@genomics.cn

The angiosperms represent one of the most diverse and species-rich groups on the planet, comprising more than 350,000 species (http://www.theplantlist.org/). There is unequivocal fossil evidence of angiosperms from the Lower Cretaceous (~132 million years ago), but molecular data indicate that they may have originated in the Jurassic or even Triassic[1,2]. Angiosperms then rapidly diversified, with a number of morphologically diverse clades appearing within a short geological timespan[1,3]. This sudden appearance of diverse angiosperm species was famously referred to by Darwin as an abominable mystery[4]. A fully resolved and well-supported phylogeny is important for understanding the origin and evolutionary history of angiosperms, but despite extensive effort, several deep-level relationships remain unclear.

Molecular data have been widely used to resolve angiosperm phylogeny for over three decades. Due to advances in sequencing technologies, declining cost, and efficient bioinformatics algorithms, we have witnessed a change in phylogenetic tree construction from a few gene regions, to whole chloroplast genomes, transcriptomes, and now large numbers of nuclear genes derived from nuclear genomes. Decades of effort have established a phylogenetic framework that identifies major angiosperm groups and clarifies their evolutionary relationships (e.g.[1,3–5], and references in supplementary Fig. 1). In this framework, Amborellales, Nymphaeales, and Austrobaileyales form a grade (ANA grade)[6,7] of successive sisters to all other living angiosperms. The vast majority (~99.95%) of angiosperms form a strongly supported clade called *Mesangiospermae* or mesangiosperms, which comprises five major clades: eudicots, monocots, magnoliids, Chloranthales, and Ceratophyllales[8]. Eudicots and monocots are the two largest clades of *Mesangiospermae*, comprising approximately 75 and 22% of species diversity;[9] magnoliids represent the third major clade with over 9000 species;[10] the remaining two groups, Chloranthales and Ceratophyllales, are small clades with only 77 and 7 species, respectively[11].

Despite extensive phylogenetic analyses and data collection efforts involving members of *Mesangiospermae*, phylogenetic relationships among the five subclades still remain ambiguous due, in part, to their rapid radiation[12], with different topologies reconstructed using various sources of morphological and/or molecular data[7] (Supplementary Fig. 1). Recent studies highlighted the conflicting placement of magnoliids as sisters to either eudicots or a clade of monocots and eudicots[5,13–21]. Analyses of nuclear genes have placed monocots as the sister to magnoliids and eudicots (with Ceratophyllales often the immediate sister to eudicots)[5,20,21]. In contrast, plastid trees yield magnoliids as the sister to a clade of monocots and eudicots, although this placement is not always well supported[1]. This incongruence between nuclear and plastid gene trees may reflect the evolutionary complexity of *Mesangiospermae*, perhaps due to incomplete lineage sorting and/or hybridization during their rapid radiation[12].

Genomic data are a powerful means for resolving phylogenetic uncertainties, and genome-scale data have been used to explore relationships within *Mesangiospermae*[21,22]. However, most of the sequenced nuclear genomes are from species representing either eudicots or monocots[23]. Nuclear genomes of *Ceratophyllum demersum*[21] and nine magnoliids (i.e., *Chimonanthus praecox*, *Chimonanthus salicifolius*, *Cinnamomum kanehirae*, *Litsea cubeba*, *Liriodendron chinense*, *Magnolia biondii*, *Piper nigrum*, *Persea americana*, and *Phoebe bournei*) have been recently published[13–19,24]. However, Chloranthales remain the last major mesangiosperm lineage without a nuclear genome assembly. This lack of data for a phylogenetically pivotal clade is a significant factor that not only impedes the resolution of angiosperm relationships, but also hinders insights into angiosperm evolution.

Chloranthales sensu APG IV[25] comprise the single family, Chloranthaceae, with four genera (*Hedyosmum, Ascarina, Sarcandra*, and *Chloranthus*) and 77 species and are considered one of the most ancient angiosperm groups based on their extensive and early fossil record[26,27]. *Chloranthus spicatus* (Thunb.) Makino is an evergreen shrub that is widely distributed in China and eastern Asia[28,29]. It is an important medicinal and horticultural plant mainly cultivated for its aromatic leaves and flowers, which are used to make a tea-like drink (Fig. 1a, b)[30], and is also a rich source of essential oils and terpenoids.

Here, we provide the nuclear genome assembly for *C. spicatus* in an effort to resolve a major enigma in angiosperm phylogeny – the relationships among the five major lineages of mesangiosperms, which represent the big bang of angiosperm evolution[12]. We also use these data to garner insights into early mesangiosperm genome evolution.

## Results

**Genome sequencing, assembly, and annotation**. *Chloranthus spicatus* has a genome size of 2.97 Gb (gigabases) based on K-mer analysis (Supplementary Fig. 2, Supplementary Data 1); the individual sequenced had a heterozygosity rate of 0.99%, which is possibly associated with the obligate outcrossing system of this species[31]. Genomic DNA was sequenced using three different methods: 182 Gb of Oxford Nanopore Technologies (ONT) long reads, 240 Gb of shotgun short reads (BGIseq 2000), and 240 Gb of Hi-C data.

The assembled genome was 2,964.14 Mb with a contig N50 size of 4.59 Mb, covering 99.7% of the genome size as estimated by K-mer analysis (Supplementary Data 2,3). Assembled contigs were then clustered into 15 pseudochromosomes, covering 99.9% of the original 2,964.14 Mb assembly, with a super-scaffold N50 of 191.37 Mb. After performing the Hi-C validation, the genome showed high contiguity, completeness, and accuracy (Supplementary Fig. 3) with the 15 pseudochromosomes corroborated by previous chromosome counts of $2n = 30$[32]. In all, 21,392 protein-coding genes were predicted using a combination of homology-based and transcriptome-based approaches. The proteome was estimated to be at least 96.8% complete based on BUSCO (benchmarking universal single-copy orthologs) assessment (Supplementary Data 4).

***Chloranthus* genome is rich in repetitive elements**. The results obtained by Tandem Repeats Finder were mapped to predict coding genes of *C. spicatus* to estimate the proportion of incorrectly detected paralogous genes (Supplementary Data 5). In the *C. spicatus* genome, repetitive elements accounted for 70.09% of the genome assembly, of which 97.67% were annotated as transposable elements (TEs) (Supplementary Data 5). Long terminal-repeat retrotransposons (LTRs) were the major class of TEs and accounted for 58.79% of the genome. Among the LTRs, the most abundant elements were *Gypsy* (68.03% of the LTRs), followed by *Copia* (19.01% of the LTRs) (Supplementary Data 6). The time of insertion of LTRs in the genome of *C. spicatus* was estimated based on a peak substitution rate of 0.03 (Supplementary Fig. 4). We assumed a synonymous substitution rate of $1.51 \times 10^{-9}$ bases per year following two recent studies of the closely related magnoliid lineages *Liriodendron* and *Chimonanthus*, resulting in an LTR burst time of approximately 9.9 Ma (see methods).

**Prevalence of longer genes and introns in Chloranthales**. Comparison of the gene and genome characteristics (e.g., genome size, gene size, exon and intron sizes) of *C. spicatus* and 17 other phylogenetically diverse flowering plants (Supplementary Data 7)

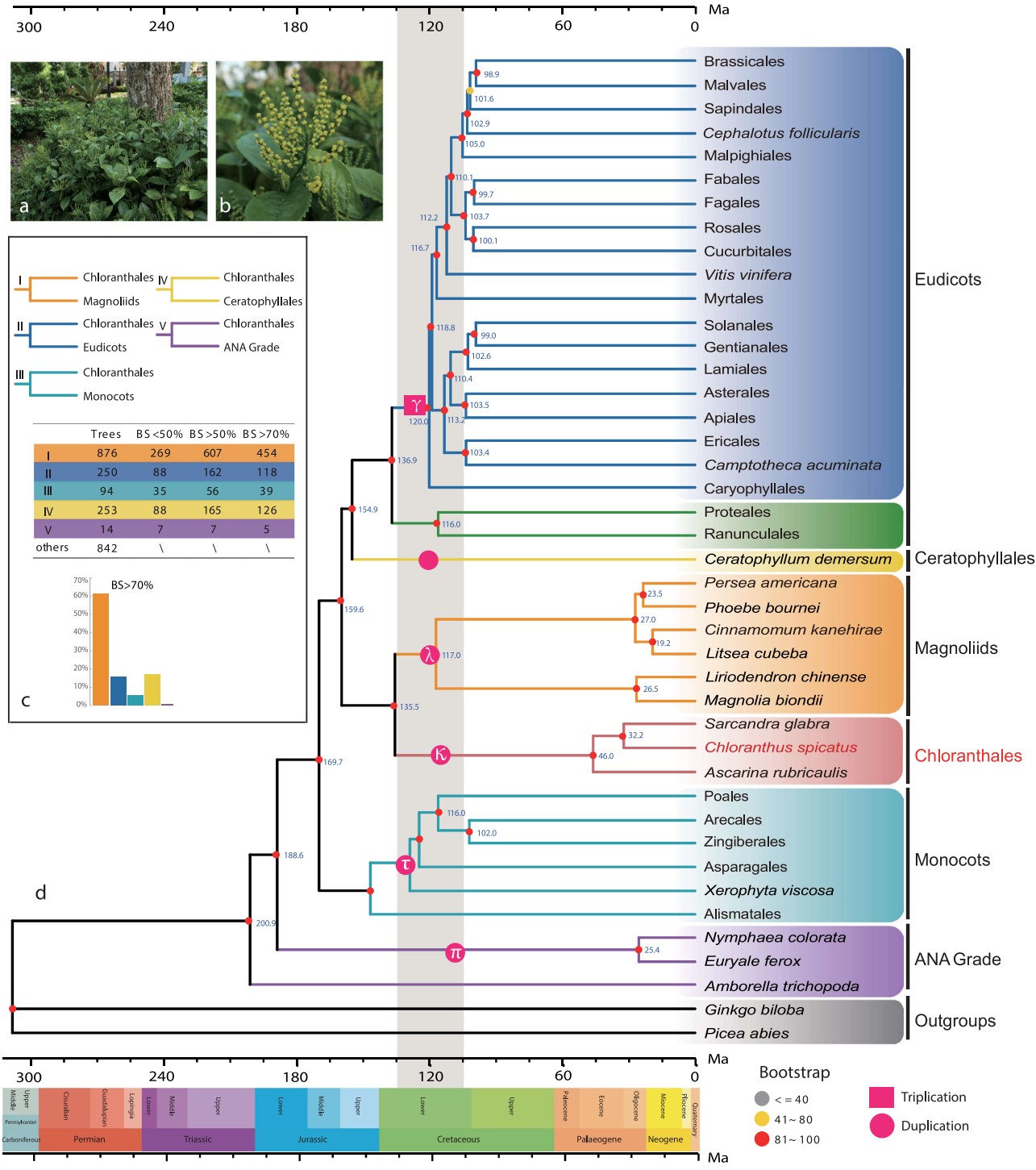

**Fig. 1 Phylogenetic summary of the flowering plants and placement of Chloranthaceae. a** *Chloranthus spicatus* in Guilin Botanical Garden, China, and **b** at the inflorescence stage (Photo credit: Qiang Zhang). **c**. A summary of phylogenetic placement of Chloranthaceae based on 2,329 low-copy nuclear (LCN) genes. In all, 61% of the gene trees placed Chloranthales as the sister lineage to magnoliids (bootstrap >70%). **d** Simplified backbone of the phylogenetic tree constructed in IQTREE using the 218 species data set, displaying the topology, polyploidy, and divergence times at the ordinal level. Whole-genome duplication/triplication events are positioned based on a previous report[33]. Source data underlying Fig. 1d are provided as a Source Data file.

revealed that long genes and long introns were more prevalent in the genomes of Chloranthales and magnoliids compared to other angiosperms (Fig. 2a, b; Supplementary Data 8). As the presence of nonfunctional genes and variation in total gene numbers among different species would bias the statistics of gene characteristics, we selected 2,184 high-confidence orthologs from *C. spicatus*, six magnoliids, and two well-characterized angiosperm

genomes, *Arabidopsis thaliana* and *Oryza sativa* (Supplementary Fig. 5a, Supplementary Data 9). Comparison of the lengths of the coding regions and introns revealed that the average coding region lengths in all nine plant genomes were similar (ranging from 1,533–1,557 bp), whereas the lengths of introns varied greatly (ranging from 153–3,681 bp; Supplementary Data 9). *Chloranthus spicatus* (3681 bp) and the six magnoliid genomes

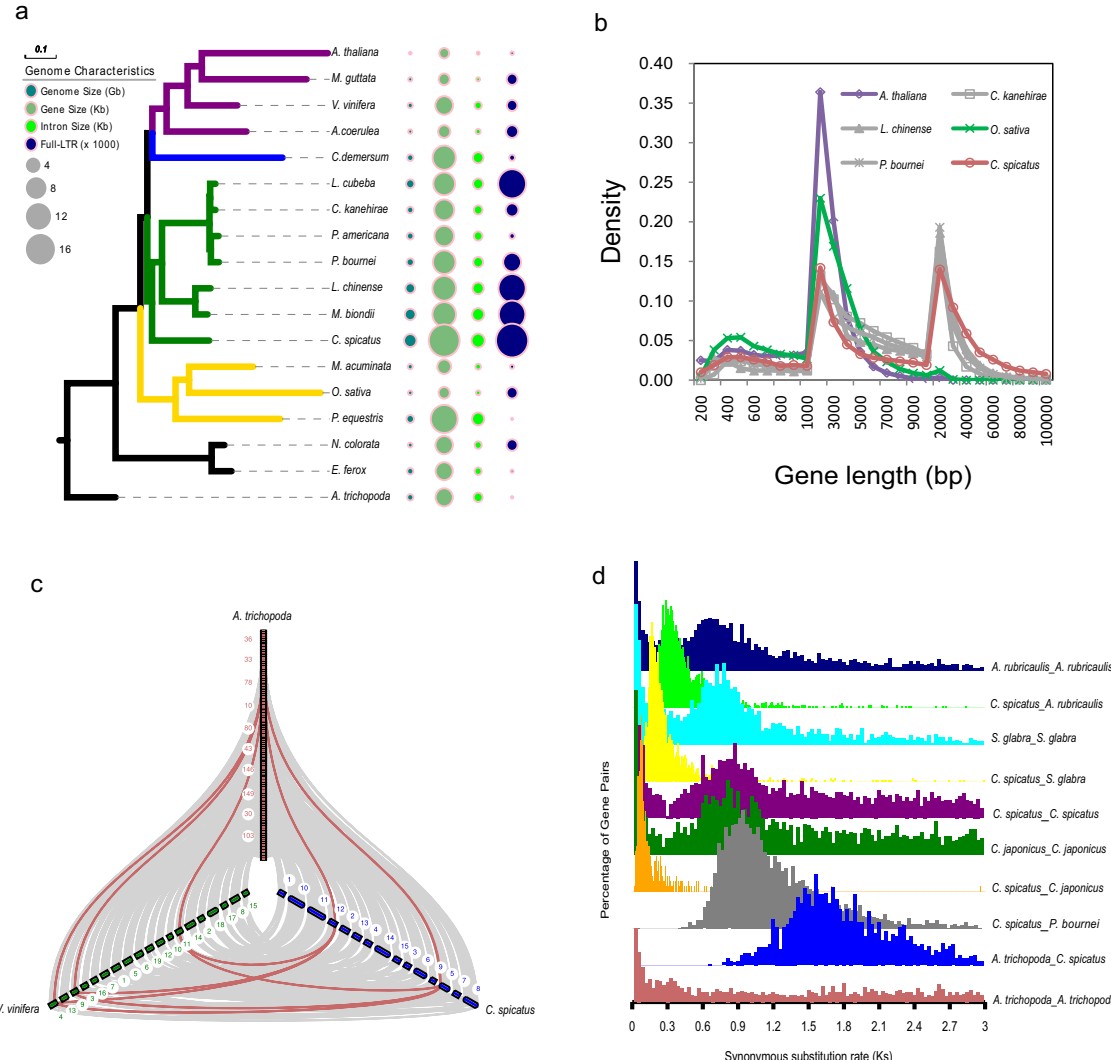

**Fig. 2 The prevalence of longer genes and introns, and confirmation of a whole-genome duplication event in *C. spicatus* based on genomic collinearity.** **a** Comparison of gene and genome characteristics (i.e., genome size, gene size, exon, and intron sizes) of *C. spicatus* and 17 other flowering plants. **b** Comparison of the lengths of the coding regions among nine representative plant genomes. **c** Collinearity patterns between genomic regions of *Amborella*, *Vitis*, and *Chloranthus*. The colored (red/grey) wedges highlight the major syntenic blocks shared among these species. **d** The number of synonymous substitutions per synonymous site (*Ks*) distributions confirming the occurrence of a whole-genome duplication (WGD) event in *C. spicatus*. Source data underlying Fig. 2a are provided as a Source Data file.

displayed much longer introns (ranging from 1,270–2,390 bp) than those of *A. thaliana* (153 bp) and *O. sativa* (372 bp), signifying that the presence of longer genes is due to the extension of the intron length rather than coding regions. We separated the 2184 high-confidence orthologs into groups based on length: <5 kb (short genes), 5–10, 10–20, and >20 kb (long genes). Long genes (>20 kb) in *C. spicatus* (876) were much greater in number than those in *Oryza* (2) and *Arabidopsis* (0) (Supplementary Data 8,9).

We found a significant correlation between intron length and genome size ($R^2 = 0.8869$). The highly conserved average length and a number of exons among the nine species compared further indicated that exon structure is mostly consistent among the angiosperms. The average length of introns was approximately 1.66 kb, 2.87 kb, and 3.35 kb for Lauraceae, Magnoliaceae, and *Chloranthus*, respectively (Supplementary Data 8).

LTR-RT represents a major fraction of plant genomes, particularly gymnosperms and magnoliids[13]. Thus, to understand

the constitution of introns in *C. spicatus*, we looked for repeated elements. LTRs were widely detected in the long introns of *C. spicatus* and appear to be the major contributor to the long introns in this species. For instance, the gene *AT1G04950.1* located on Chromosome 1: 1402606–1408184 encodes Transcription initiation factor TFIID subunit 6 in *A. thaliana*. The LTR length of this orthologous gene in *C. spicatus* (Cspi02386) was significantly longer than that in Lauraceae, Magnoliaceae, *O. sativa*, and *A. thaliana* (Supplementary Fig. 5b).

We discovered 11,500 intact LTRs and classified them into two groups, Gene-20K LTR (LTRs distributed in genes >20 kb length) and ALL LTR (LTRs distributed throughout the genome, Supplementary Fig. 6). A similar model distribution of Gene-20K LTR and ALL LTR (Supplementary Figs. 7, 8) suggested that the insertion timing of both LTR groups was the same. Further analyses of expression levels revealed that genes with short introns were more likely to exhibit low expression, while genes with long introns exhibited higher expression. However, when the

intron length was larger than 40 kb, the expression level subsequently declined in *C. spicatus* (Supplementary Fig. 9).

**Conserved synteny and ancient WGD.** Our investigation of collinearity and synteny patterns between genomic regions of *Amborella trichopoda* (sister to all other extant angiosperms), *Vitis vinifera* (sister to all other rosids), and *C. spicatus* showed highly conserved synteny among these three species (Fig. 2c). In addition, this analysis showed clear structural evidence of a single ancient WGD in *C. spicatus*. The syntenic depth ratio between *C. spicatus* and *A. trichopoda* was found to be 1:2, which means that each *A. trichopoda* region could be matched to two genomic regions in the *C. spicatus* genome while the comparison of *C. spicatus* with the ancient hexaploid *V. vinifera* genome revealed a 2:3 syntenic depth ratio (Fig. 2c).

To further investigate the extent of conservation of genome structure between *C. spicatus* and other angiosperms, we performed pairwise synteny comparisons with several species of magnoliids (*Magnolia biondii, Liriodendron chinensis, Persea americana, Cinnamomum kanehirae, Litsea cubeba, Phoebe bournei*) (Supplementary Data 10). Our results clearly showed that *C. spicatus* shared a higher number (3,029; i.e., 62.7%) of syntenic blocks (at both the scaffold and chromosome level) with species in its sister clade, the magnoliids, than with Ceratophyllales (2,483, 52.5%), *V. vinifera* (2,275, 56.5%), or the monocot *Oryza sativa* (1,700, 45.3%) (Supplementary Fig. 10, Supplementary Data 10). *Amborella trichopoda* (1,150, 57%) shared the fewest syntenic blocks with *C. spicatus* among all the species used for comparative analysis (Supplementary Data 10); overall, the number of shared syntenic blocks between these representative genomes generally coincided with phylogenetic relationships.

To further investigate the phylogenetic placement of the *C. spicatus* WGD, we compared the distribution of $K_s$ values, the number of synonymous substitutions per synonymous site. The $K_s$ distribution for *C. spicatus* paralogs showed an obvious peak at approximately $K_s = 0.9$, and peaks at similar $K_S$ values were identified for other species (*Ascarina rubricaulis, Chloranthus japonicus*, and *Sarcandra glabra*) of Chloranthales (Fig. 2d); the coincidence of these $K_S$ values suggests that an ancient WGD event may be shared among all extant members of this clade. Further, the $K_S$ values for orthologs shared by *C. spicatus* and *Phoebe bournei* (Lauraceae; magnoliids) show a peak value slightly greater than that observed for paralogs in *C. spicatus* and other Chloranthales species, which suggests that the Chloranthales WGD occurred after the divergence of Chloranthales and magnoliids (Fig. 2d). These observations suggest that the ancient WGD event we detected was specific to Chloranthales.

Although magnoliids also exhibit an ancient WGD event (Supplementary Data 11a), this event was not shared with Chloranthales. The age of the Chloranthales WGD is similar to that of a number of ancient polyploidy events that occurred independently in several major clades of angiosperms: the gamma (γ) event (103.67–129.35 Ma) in the common ancestor of core eudicots; the tau (τ) event (101.82–138.82 Ma) during the early diversification of monocots; the lambda (λ) event (98.22–130.04 Ma) during the early diversification of magnoliids; the pi (π) event (85.78–119.82 Ma) in Nymphaeales; the kappa (κ) event (98.06–130.54 Ma) in Chloranthales (this study); and an unnamed event specific to Ceratophyllales[33] (Fig. 1d). Although these WGD events occurred independently, many of the same stress-related genes were retained independently after these WGD events, including two heat shock transcription factors and *Arabidopsis* response regulators[34]. These genes also appear to be retained in *Chloranthus* (Supplementary Figs. 11, 12).

**Phylogenomic placement of Chloranthales as sister to magnoliids.** To resolve the long-standing uncertainty regarding the phylogenetic position of Chloranthales and relationships among the five major lineages of *Mesangiospermae*, 257 single-copy nuclear (SCN) genes were identified using whole-genome sequences from *C. spicatus* and 17 other flowering plants (strict single-copy genes for each species without missing genes; see Methods for species). The aligned protein-coding regions were analyzed using coalescent and concatenation approaches. Both analyses yielded an identical and highly supported topology (bootstrap values of 100%) (Supplementary Fig. 13) in which monocots were sister to all other mesangiosperms; Chloranthales appeared as the sister group to magnoliids, and Chloranthales + magnoliids together were sister to Ceratophyllales + eudicots (Fig. 2a, Supplementary Fig. 13). We also performed phylogenetic inference based on a 937-SCN gene data set with selection criteria allowing a maximum of three missing species. The phylogenetic results showed an identical topology to that of the 257-SCN gene data set, supporting Chloranthales as the sister to magnoliids (Supplementary Fig. 13).

To avoid potential errors caused by sparse gene sampling, we extracted 2,329 low-copy nuclear (LCN) genes, allowing a maximum of three copies for each gene in each species. The phylogenetic trees were then similarly reconstructed by both coalescent and concatenation methods as described above. The resulting species trees were topologically identical to the phylogenetic findings as revealed above based on the 257-SCN and 937-SCN data sets (Supplementary Fig. 13). Among the 2,329 LCN gene trees, 61% of the trees (454 out of 742 trees) placed Chloranthales as the sister lineage to magnoliids, with bootstrap support greater than 70% (type I, Fig. 1c).

As poor taxon sampling may lead to topological errors, we added a large number of published genomes of the angiosperms and two transcriptomes of Chloranthales to increase our taxon sampling. We extracted 612 'mostly' single-copy orthologous genes following Yang et al.[21] and generated a 218-species dataset. The phylogenetic results were congruent with the topologies based on analyses of the 257-SCN, 937-SCN, and 2,329-LCN data sets, supporting monocots and a clade of Chloranthales plus magnoliids as successive sister lineages to a clade of Ceratophyllales plus eudicots (Fig. 1d, Supplementary Fig. 14).

Phylogenetic analyses were also conducted based on chloroplast DNA sequence data. We selected 80 genes, following a recent study that analyzed 2,881 plastomes[1], and obtained two data sets, with 18 species and 134 species, respectively. The resultant topologies using both chloroplast data sets agree with those from the four nuclear data sets in strongly supporting a sister relationship between Chloranthales and magnoliids (Supplementary Figs. 15, 16).

**Gene tree incongruence in *Mesangiospermae*.** Although the same pattern of phylogenetic relationships among the five major groups of *Mesangiospermae* was consistently recovered in analyses of all four nuclear data sets, phylogenetic incongruence was observed among nuclear gene trees. A major conflict was identified among single-gene trees in all four nuclear gene data sets (257-SCN, 937-SCN, 2,329-LCN, and 612-SCN) involving the placement of the Chloranthales-magnoliids clade relative to monocots and eudicots. We summarized the conflict among gene trees in the 2,329-LCN data set with regard to the proportions of trees supporting three different branching patterns for Chloranthales-magnoliids, monocots, and eudicots. The percentage of gene trees supporting the Chloranthales-magnoliids clade plus eudicots together forming a sister group to monocots (Type II) was higher than percentages for the other two topologies (gene

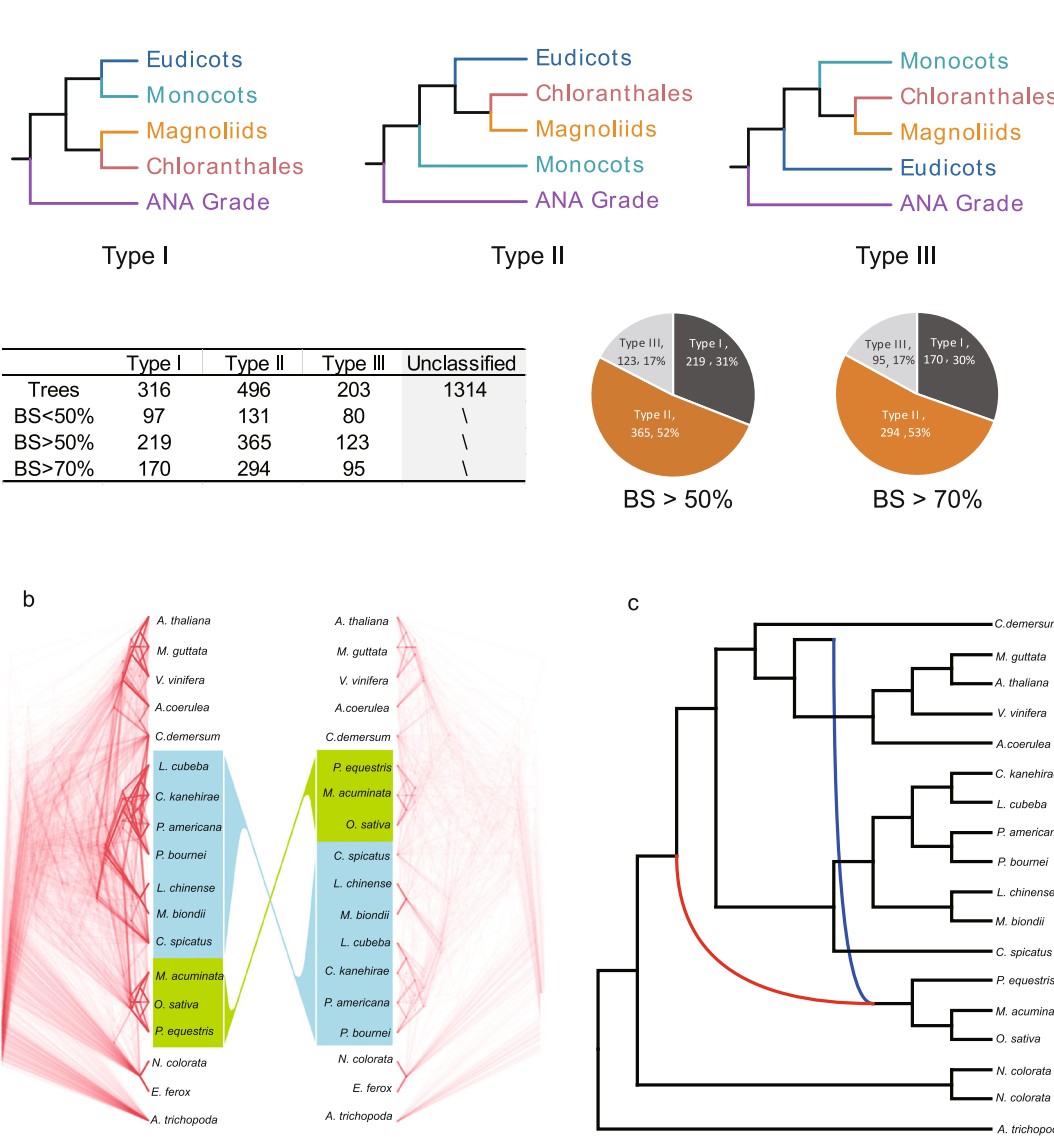

**Fig. 3 Gene tree discordances among the five major groups of *Mesangiospermae*. a** A summary of the conflicts among gene trees in the 2,329-LCN data set with regard to the proportions of trees supporting three different branching patterns for Chloranthales-magnoliids, monocots, and eudicots. **b** Gene tree incongruence between nuclear (2,329 LCN genes) and plastid (80 plastid genes) trees in a consensus DensiTree plot. **c** A consensus scenario showing ancient gene flow between monocots and eudicots, inferred by QuIBL, PhyloNet, and ABBA-BABA D-statistics. Source data underlying Fig. 3b are provided as a Source Data file.

trees with BS > 70%: Type I, 30%; Type II, 53%; Type III, 17%; Fig. 3a). It is notable that Type I and Type III, the two relationships conflicting with the most likely species tree, are not equal in frequency, suggesting gene tree incongruence patterns not expected under ILS alone (below).

Furthermore, gene tree discordances were also observed between chloroplast and nuclear gene trees. Phylogenetic analyses of these 18 and 134 flowering plants inferred from 80 concatenated plastid genes strongly supported the placement of the Chloranthales-magnoliids clade as sister to all other *Mesangiospermae* (Fig. 3b and Supplementary Figs. 15, 16), which is consistent with the Type I nuclear topology (Fig. 3a).

A nonrandom incongruence pattern was observed among different topology types: constituent species of monocots (3 spp.), eudicots (4 spp.), and magnoliids (7 spp.) were assigned to a clade.

For each topology type, the majority of genes supported the monophyly of *C. spicatus* and seven species of magnoliids (Type I: 168/316 = 53.2%; Type II: 297/496 = 59.9%; Type III: 122/203 = 60.1%). We also mapped genes that caused conflict on the chromosomes. Genes that supported both Type I and Type II topologies were found to be evenly distributed on the 15 chromosomes (Supplementary Fig. 17). Chi-squared tests showed that gene numbers and locations on each chromosome do not differ significantly (Supplementary Data 12, 13).

**Distinguishing hybridization from incomplete lineage sorting.** The observed gene tree incongruence between nuclear and chloroplast trees and among nuclear single-gene trees indicates the possibility of incomplete lineage sorting (ILS) and/or

hybridization events during early angiosperm evolution. We first used QuIBL, an approach using branch length distributions across gene trees, to infer putative hybridization patterns[35]. In all, 100 runs of QuIBL were conducted using 500 randomly selected trees from the 2,329 LCN gene trees. Strong hybridization signals (rate >0.1) were identified in several pairs of major clades of angiosperms (Supplementary Figs. 18, 19), including: (i) ancestor of eudicots and ancestor of monocots; (ii) ancestor of eudicots and *C. spicatus*; (iii) ancestor of the species pair *Arabidopsis thaliana-Erythranthe guttata* and *Vitis vinifera*; (iv) *Erythranthe guttata* and *Ceratophyllum demersum*. Strong signals of ILS were also detected between Lauraceae and Magnoliaceae (Supplementary Figs. 18, 19). Among these events, cases (i) and (ii) can be explained as the causes of gene tree incongruence of the Chloranthales-magnoliids clade relative to monocots and eudicots.

A second analytical approach, PhyloNet, was used to further assess putative hybridization events in our phylogeny. Five network searches were carried out by allowing one to five reticulation events. The species network under the best model (AICs = 50.78; BICs = 30.52, Supplementary Data 14) identified two hybridization events among major clades of angiosperms (Supplementary Fig. 20a), supporting ancient hybridization between early members of Nymphaeales and monocots. Ancestral gene flow between monocots and eudicots (Supplementary Fig. 20a) was additionally supported by results of QuIBL (Supplementary Fig. 19). To test whether the PhyloNet results identified hybridization correctly, we repeated the PhyloNet analyses using coalescent trees simulated without hybridization under the ASTRAL species tree (Supplementary Fig. 11). As expected, the species network under the best model (AICs = 51.21; BICs = 30.25, Supplementary Data 14) detected no hybridization events among monocots, eudicots, and magnoliids (Supplementary Fig. 20b), suggesting that the analysis using empirical gene trees was not susceptible to false positives.

The unequal frequencies of Type I and Type III topologies discordant with the species tree suggested that ILS alone may not explain the gene tree conflicts in this study; therefore, we also used the ABBA-BABA approach to explicitly model patterns of discordant genealogies. This analysis also inferred frequent hybridization signals (Supplementary Fig. 21). Consistent with the other two methods employed, the hybridization event between monocots and eudicots was detected, with the largest absolute Z-value (14.6).

In summary, all three methods used to investigate hybridization (QuIBL, PhyloNet, and ABBA-BABA D-statistics) were unanimous in suggesting ancient gene flow between monocots and eudicots, although with variation among methods in the number of hybridization events and any additional lineages involved in hybridization. A consensus scenario is presented (Fig. 3c) showing ancient gene flow between monocots and eudicots.

**Evolution and expansion of terpene synthase genes**. Terpene synthases (TPSs) are key enzymes that control the production of terpenoids, crucial defense compounds in plants[36]. To explore the evolution of the TPS family in *Magnolia* and Chloranthales, as well as to garner a better understanding of terpene evolution in angiosperms, we searched for candidate TPSs in *C. spicatus* and 17 other flowering plants (the same taxon sampling as in the comparative genomics analyses). *Chloranthus spicatus* encodes 73 TPSs (Supplementary Data 15), similar to *V. vinifera* (75) and *A. coerulea* (74), while *C. kanehirae* exhibited the largest number (95) of TPSs. Particularly, compared to the ANA grade, there was higher diversity in almost all of the magnoliid species and *C. spicatus* (Supplementary Data 16). Furthermore, according to

the subfamily classification of TPS genes, TPSs were divided into 6 clades: TPS-a, b, c, e, f, and g (Fig. 4b). In *Amborella* and members of Nymphaeaceae (*Euryale ferox* and *Nymphaea colorata*), TPS-a was absent (Supplementary Data 16). Furthermore, when we performed GO enrichment analyses using the shared genes between magnoliids and Chloranthales, the genes related to terpene synthase activity (GO:0010333) exhibited a low *P*-value, indicating that terpene synthase activity was the most enriched of all GO categories (Supplementary Data 17). Moreover, our gene family analysis indicates that the TPS-a and TPS-b gene clades expanded remarkably in magnoliids and Chloranthales compared to all other angiosperm clades (Supplementary Data 16, Fig. 4b); these gene clades primarily consist of angiosperm-specific sesquiterpene and monoterpene synthases, respectively. Several unique *Chloranthus*-specific sesquiterpenoids, including chlorahololides A, chloranthalactone A, and chlotrichenes A and B with bioactive potential, have been isolated and chemically synthesized in the lab[37–39].

To understand the origin of paralog generation of these TPS genes, we compared the numbers of genes in each duplication type among species of magnoliids and *Chloranthus* (Supplementary Data 18). The results showed tandem (23, 33.3%), WGD (18, 26.1%), and transposition (21, 30.4%) duplication events contributed to the expansion of TPSs in *C. spicatus*, with only a few proximal repeats (7, 10.1%). The 73 CsTPS genes are not evenly distributed across the 15 chromosomes, with Chr2 having the highest concentration of TPS genes. Tandem repeats are mainly present on Chr2 and Chr7 (5 and 6 tandem repeats, respectively), but are also present on chromosomes 4, 5, 9, 14, and 15. We hypothesize that WGD contributed to TPS expansion as well, for instance, the higher copy number of the pairs CsTPS03 and CsTPS33 and CsTPS05 and CsTPS19 (Supplementary Fig. 22, Supplementary Data 16, 18).

Next, we investigated the genes involved in the production of non-volatile isoprenoids via the 2-C-methyl-D-erythritol 4-phosphate (MEP) pathway and the mevalonate (MVA) pathway and identified 44 genes in *C. spicatus* that may be involved in these pathways (Supplementary Data 19, 20). There were multiple copies of the genes encoding enzymes related to the MVA pathway, and the number was approximately double that detected for genes in the MEP pathway. The gene encoding the HMGR enzyme (Hydroxy-3-methylglutaryl) displayed the highest number of gene copies (12) followed by AACT (Acetoacetyl-CoA thiolase) (6 copies). In the MEP pathway, except for DXS (1-deoxy-D-xylulose-5-phosphate synthase), DXR (1-deoxy-D-xylulose 5-phosphate reductoisomerase1-deoxy-D-xylulose 5-phosphate reductoisomerase), and GGPS (geranylgeranyl pyrophosphate synthase), each remaining enzyme had only one corresponding gene copy. In addition, to further validate this observation, a differential gene expression (DE) analysis was also performed using the transcriptome data from different plant parts (stamen, carpel, and peduncle) (Fig. 4a). Regardless of the number of gene copies encoding the enzymes of these pathways, at least one gene copy for each enzyme was highly expressed in each tissue. However, for the multiple-copy genes, a few genes were responsible for most of the expression, while the remaining copies were weakly expressed. Altogether, the analyses of expansion and differential expression of TPSs suggest that the appearance of multiple-copy genes in the MVA pathway could be related to the expansion of the TPS-a subfamily, which is probably responsible for the production of sesquiterpenes in Chloranthales.

**Distribution of *R* genes in *Chloranthus***. Nucleotide-binding site-leucine-rich repeat (NBS-LRR, NBS for short) genes

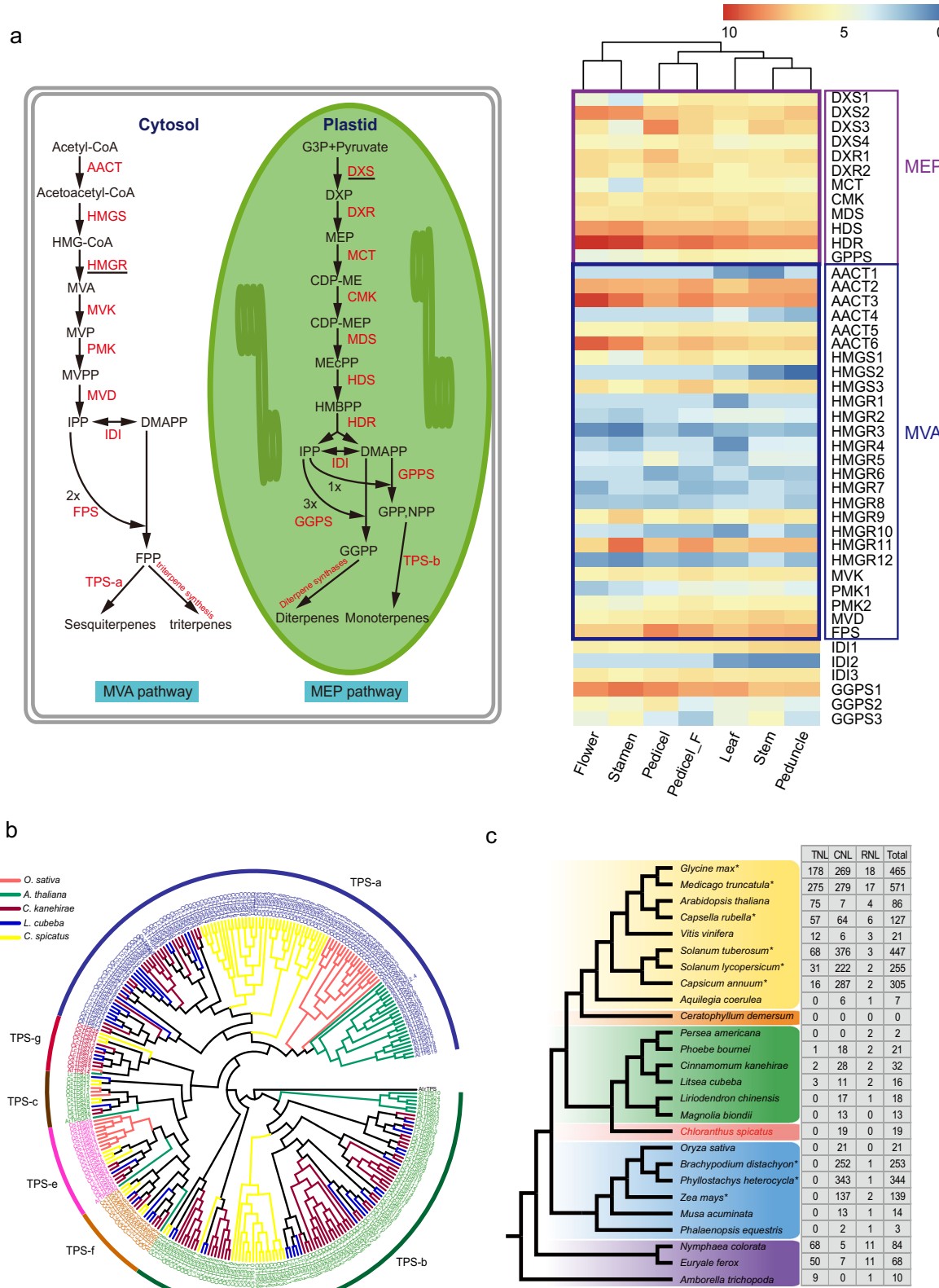

encompass more than 80% of the characterized *R* genes[40]. The NBS genes were divided into three classes, namely, TIR-NBS-LRR (TNL), CC-NBS-LRR (CNL), and RPW8-NBS-LRR (RNL)[40]. We identified 3,518 nucleotide-binding site-leucine-rich repeats (NBS-LRR, NBS for short) genes in 28 angiosperm species, and the nucleotide-binding site-leucine-rich repeat (NBS-LRR) genes were classified into three classes: the Toll/interleukin-1 receptor

TIR-NBS-LRR (TNL), N-terminal coiled-coil motif CC-NBS-LRR (CNL), and resistance to powdery mildew8 RPW8-NBS-LRR (RNL)[40] (Supplementary Data 21). The gene copy number in each NBS class showed considerable variation among the analyzed species (Fig. 4c). The highest number of TNL genes was found in *M. truncatula* (of the 28 species examined), while the highest number of CNLs and RNLs were in *S. tuberosum* and

**Fig. 4 Evolution and expansion of terpene synthase genes and contraction of *R* genes in Chloranthales. a** A total of 44 genes related to the 2-C-methyl-D-erythritol 4-phosphate (MEP) pathway and the mevalonate (MVA) pathway were identified in *C. spicatus* (left panel). HMGR and DXS exhibited the highest copy numbers in the MEP and MVA pathways, respectively. Differentially expressed genes among seven representative tissues of *C. spicatus* involved in MEP and MVA pathways are shown in the right panel. **b** Identification of candidate terpene synthases (TPSs) in *C. spicatus* and subfamily classification revealed six major clades (TPS-a, b, c, e, f, and g). The gene family tree indicates that TPS-a and TPS-b gene clades are significantly expanded in magnoliids and Chloranthales. **c** Contraction of *R* genes in Chloranthales. The nucleotide-binding site-leucine-rich repeat (NBS-LRR) genes were divided into three classes: TIR-NBS-LRR (TNL), CC-NBS-LRR (CNL), and RPW8-NBS-LRR (RNL). In all, 3,518 NBS genes were identified in 28 angiosperm species. '*' indicates the data were obtained from a previous study[40].

*G. max*, respectively. Moreover, *M. truncatula*, *G. max*, and *S. tuberosum* contained more *R* genes than the other angiosperms examined; Chloranthales and magnoliids contained many fewer *R* genes. TNL and RNL were absent from *Chloranthus* and the magnoliids (as in the monocot species, *O. sativa*), and only 19 and 13 CNLs were present in *C. spicatus* and *Magnolia biondii*, respectively (Fig. 4c, Supplementary Data 21). In the species having both TNL and CNL genes, the CNLs are generally more common than the TNLs, with the exception of *A. thaliana*, *V. vinifera*, *A. trichopoda*, *E. ferox*, and *N. colorata*.

## Discussion

We assembled a high-quality chromosome-level genome of *Chloranthus spicatus* by combining long-read Nanopore sequences with highly accurate short reads from BGI-DIPSEQ sequencing and using Hi-C data for super-scaffolding. As the representative genome of Chloranthales, it fills a key evolutionary gap on the tree of life. The availability of the *Chloranthus* genome contributes to a better understanding of deep angiosperm diversification and phylogeny and to patterns of genome evolution in angiosperms. This assembly also facilitated an in-depth comparative analysis of genome evolution in *Chloranthus* and species of magnoliids, the sister group of Chloranthales.

WGD events have been considered a major factor driving the evolution and diversification of angiosperms[5,33,41] and may have contributed to the evolution of key innovations and adaptation to diverse environments[42–44]. We confirmed the occurrence of an ancient WGD (κ) prior to the diversification of extant Chloranthales.

The phylogenetic position of Chloranthales within *Mesangiospermae* has long been uncertain based on various sources of molecular data (Supplementary Fig. 1). In this study, we performed phylogenetic inference using four nuclear data sets (257-SCN, 937-SCN, and 2,329-LCN from 18 genomes, 612-LCN genes from 216 genomes and two transcriptomes), and two chloroplast data sets (80 genes from 18 and 134 species). All analyses yielded an identical highly supported sister-group relationship between Chloranthales and magnoliids. However, the relationship of this Chloranthales + magnoliid clade differed among trees from these six data sets and among genes within data sets. One recurring aspect of recently published genomic studies is their conflicting phylogenetic placement of magnoliids relative to monocots and eudicots. Some analyses based on nuclear genes found magnoliids sister to eudicots whereas others placed them sister to a clade of monocots and eudicots[13,14,16–19,24]. Incongruent relationships of magnoliids were not only identified in different genome papers based on nuclear data, but were also detected between nuclear and chloroplast data sets[17,21]. Large amounts of plastid sequence data, including our own here, have placed Chloranthaceae sister to magnoliids, with that clade in turn sister to monocots + (Ceratophyllaceae + eudicots). In contrast, concatenated nuclear genes typically place Chloranthaceae + magnoliids as sister to Ceratophyllaceae + eudicots, with monocots sister to this entire clade. Many factors could be

responsible for these topological conflicts such as taxon sampling, incomplete lineage sorting, or ancient hybridization[23]. Using large taxon sampling (218 species in our nuclear data set and 134 species in the chloroplast data set), we identified the same pattern of topological conflict between nuclear and chloroplast gene trees. The results suggest that limited taxon sampling alone does not explain the observed gene tree incongruence, highlighting the necessity of identifying alternative causes such as incomplete lineage sorting or hybridization.

Deep phylogenetic incongruence between nuclear and organellar genomes has been recently reported in angiosperms, and incomplete lineage sorting has been proposed as one possible cause for the incongruence[1,7,16,17,21]. However, distinguishing between the two hypotheses of incomplete lineage sorting and ancient hybridization is difficult, as these evolutionary processes can result in similar phylogenetic patterns[45], although with alternative gene tree distribution predictions[46]. Previous genomic analyses that examined the conflicting position of magnoliids, did not fully evaluate the underlying causes[21]. Yang *et al.* performed PhyloNetworks and coalescent simulation analyses and suggested that incomplete lineage sorting may account for the incongruent phylogenetic placement of magnoliids between nuclear and plastid genomes[21], which differs from our conclusion suggesting ancient reticulation between eudicots and monocots (Supplementary Fig. 20). An ancient gene flow event between monocots and eudicots was detected by all three methods (QuIBL, PhyloNet, and ABBA-BABA D-statistics) used in this study; simulations demonstrated the results were robust to potential false positives. Our study suggests ancient gene flow is more likely than incomplete lineage sorting as the cause of the incongruent phylogenetic placement of magnoliids. The key role of ancient hybridization revealed here sheds additional light on the difficulty of inferring the branching order of major angiosperm lineages, traditionally explained by short divergence times alone. Our findings also raise important questions about the relative roles of different diversification mechanisms for the early explosion of angiosperms.

Our analyses revealed notable TPS gene family expansions in Chloranthales and magnoliids compared to members of the ANA grade. We found considerable expansion of P450 gene families such as CYP71B, CYP84, CYP706, CYP78, CYP79A, CYP72, CYP719 in *C. spicatus*, suggesting their plausible role in functionalization of the TPSs backbone[47] (Supplementary Fig. 23). Based on previous studies, most terpenoid-related cytochromes P450 are members of the CYP71 clade, and are involved in the formation of the sesquiterpene ester ring[37,48]. We also discovered a relatively high number of copies in the CYP71 gene family in *C. spicatus* in comparison to its magnoliid relative, *L. chinense* (Supplementary Fig. 23). The significant expansion of TPS family genes, particularly TPS-a and TPS-b genes, may be responsible for the rich volatile content of magnoliids and Chloranthales[13,17,49]. Likewise, by exploring available angiosperm genome sequence data, we were able to identify several sets of NBS-encoding genes and identify possible patterns of evolution of these *R* gene subclasses during angiosperm evolution. CNL

subclass genes are the most common type of *R* genes across angiosperms (except in Brassicaceae[50]). CNL genes expanded in both monocots and eudicots;[40] based on the nuclear phylogeny found here, these expansions were independent events that would have occurred in parallel. The presence of only CNL genes, and the absence of both TNL and RNL genes in Chloranthales, magnoliids, and monocots, suggests that *R* genes diversified in the eudicots. However, the presence of TNLs in *Amborella* and *Nymphaea* suggests a more complicated pattern of evolution of this subclass than RNLs.

## Methods

**Sample preparation, library construction, and sequencing.** The genomic DNA of *C. spicatus* (Voucher number: GX20200316-1, SCBG) was extracted from a specimen collected at Fairy Lake Botanical Garden, Shenzhen, China. For genome sequencing, the DNA was isolated from the leaves of *C. spicatus* by utilizing the CTAB (cetyltrimethylammonium bromide) protocol[51]. Next, the sequencing of the library was performed using BGI-DIPSEQ[52], which generated ~374.45 Gb of 100-bp paired-end data having ~250 bp insert size (Supplementary Data 1). The raw reads with adaptor contamination and duplicated reads were filtered with Trimmomatic (v0.40)[53].

The ONT library[54] was built with the LSK108 kit (SQK-LSK108, Oxford) and sequenced on the Nanopore GridION X5 sequencer utilizing five flow cells[55]. With the help of Albacore and the MinKNOW package (v 4.0.4), base calling was carried out, which resulted in a total of ~9.09 million Nanopore reads, or ~182.48 Gb raw data with an N50 of ~20.79 kb (Supplementary Data 1, 2).

The Hi-C libraries were constructed[56] at BGI Qingdao Institute. DNA from young leaves of *C. spicatus* were subjected for digestion using *Mbo*I in accordance with the standard Hi-C library preparation protocol. The Hi-C libraries were sequenced on the BGI-DIPSEQ platform, yielding ~251.80 Gb of data with 100-bp paired-end reads (Supplementary Data 2).

Young leaf, stem, and pedicel tissues of *C. spicatus* were collected for transcriptome sequencing. We used the TIANGEN Kit to extract total RNA and the NEBNext UltraTM RNA Library Prep Kit to prepare paired-end libraries with an insert size of 250 bp. Libraries were barcoded and pooled together as input to the BGI-DIPSEQ platform for sequencing, resulting in an average of 9.27 Gb sequences on the BGI-DIPSEQ platform (Supplementary Data 1), to ensure complete coverage for each transcriptome. For further RNA-sequencing analysis, 6 Gb of 100-bp paired-end data for each tissue was used after the removal of low-quality data.

**Genome size estimation.** According to earlier reports[57], *C. spicatus* has a genome size of ~3.0 Gb based on flow cytometry and possesses 15 chromosome pairs ($2n = 30$)[57]. In this study, the genome size of *C. spicatus* was estimated using K-mer frequencies[58]. To minimize the sequencing error rate, strict quality control with Trimmomatic (v0.40)[53] was followed by filtering out: reads with >5% of "N"; low-quality reads with >25% of their bases as low-quality; and PCR duplicates. Based on the results of the 17-mer frequency distribution analysis (Supplementary Fig. 2) with GenomeScope[59], we estimated the genome size of *C. spicatus* to be 2.97 Gb, which is very close to previous estimates by flow cytometry. In addition, a secondary peak was also observed, suggesting noticeable heterozygosity (~1%), which presented considerable difficulties in performing the genome assembly (Supplementary Fig. 2).

**Genome assembly and assessment of assembly quality.** De novo assembly of the raw ONT long reads was performed using NextDenovo assembler (v2.2) with the default parameters read_cutoff = 1 k, seed_cutoff = 25,211, and NextPolish (v1.3.0)[60] was used six times [ONT long reads (two rounds) and short reads (four rounds)] to polish the initial draft assembled contigs. Because this genome had a high heterozygosity rate, we used purge_dups (v1.2.3) to remove primary contigs that had both haplotypes rather than a contig and its associated haplotig. This tool considers the mapped read coverage with short read and Minimap2 alignments[61] and screens the contigs for the assembly of the haploid genome. Following that, contigs were searched against the NCBI non-redundant bacterial database using the BLAST algorithm (v2.2.29)[62] to rule out any potential bacterial contamination. Contigs with an identity greater than 90% and an alignment of at least 80% were excluded. The final assembly had a total length of 3.03 Gb (N50: 5.78 Mb) spread across 117 contigs (Supplementary Data 2).

Evaluation of the *C. spicatus* genome assembly is provided by the following. First, mapping of the 1,375 conserved core eukaryotic genes from the BUSCO data set (embryophyta_odb10, BUSCO v3.0.2)[63] resulted in 89.20% (Supplementary Data 4) of the core eukaryote genes recovered for the majority of the genome assembly. Second, to evaluate the DNA read mapping rate (>98.3%), the reads were mapped to the draft assembly using BWA (v.2.21)[64] along with genome coverage (>98.5%), and GC-depth distribution (Supplementary Data 2, 3). Third, we used BLAT (v.36)[65] to compare the identity and coverage (Supplementary Data 1) between the draft assembly and the transcriptome assembled by Trinity (v2.6.6)[66]

(see below). Finally, we mapped the RNA reads to the draft assembly to assess the read rate of RNA mapping with Hisat2[67] (Supplementary Data 1).

**Chromosome assignment using Hi-C.** Trimmomatic (v0.40)[53] was used to trim high-quality paired-end reads to remove low-quality bases and adapter sequences from the reads, and all filtered reads were aligned to contigs using a juicer (https://github.com/aidenlab/juicer, v3)[68] to calculate the contact frequency. Then, 3ddna (v180922)[69] was used with two iterative rounds for misjoin correction (-r2) using default parameters. The oriented scaffolds were utilized to generate the interaction matrices with a juicer to inspect and manually correct with Juicebox assembly tools (v1.11.08)[68].

**Annotation of repetitive elements.** With the help of the software Tandem Repeats Finder (4.07)[70], we were able to detect tandem repeats throughout the genome. A combination of homology-based comparisons with RepeatMasker (4.0.5)[71] and de novo approaches with LTR_retriever[72], LTR_FINDER (1.0.6)[73], RepeatModeler2[74], and MITE-hunter[75] was used to predict the Transposable elements (TEs). The MITE, LTR, and TRIM (Terminal repeat Retrotransposon In Miniature) repetitive sequence libraries were combined to generate a complete and non-redundant custom library. The repeat library was further used as an input in RepeatMasker to categorize transposable elements.

Regions of LTR-retrotransposon sequences coding for reverse transcriptase (RT) and integrase (INT) protein domains were identified using DANTE-Protein Domain Finder (https://github.com/kavonrtep/dante/), a tool available at the RepeatExplorer server[76], which employs LAST searches against a custom database of transposon protein domains. The hits were filtered to ensure that they covered at least 80% (-thl) of the reference sequence, with a minimum identity of 35% (-thi) and a minimum similarity of 45% (-ths), with a maximum of three interruptions allowed per sequence (frameshifts or stop codons). The relative amounts of the different repetitive element classes, orders, and families are given in Supplementary Data 5 and 6.

The whole genome was searched for the characteristic structure of full-length retrotransposons (full-LTRs) using LTR_retriever[72]. We used the EMBOSS (6.5.7.0) package distmat to calculate the K-value (the average number of substitutions per aligned site), and used the KaKs_Calculator (v2.0) software to calculate the Ks value of the retrotransposons' 5'- and 3'-LTR sequences, which were then used to estimate insertion times using equation 1:

$$T = Ks/(2 * r) \tag{1}$$

where r is the average substitution rate of $1.51 \times 10^{-9}$ substitutions per year per synonymous site[77].

**Protein-coding gene prediction and functional annotation.** The protein-coding gene set of *C. spicatus* was deduced by de novo, homology, and evidence-based gene prediction (transcriptome data). De novo gene prediction was evaluated using various software including Augustus (3.0.3)[78], GlimmerHMM (3.0.1)[79], and SNAP (version 11/29/2013)[80] on a repeat-masked genome. An initial subset of the transcriptomic data was used to generate the training models. Homologous gene prediction was performed by comparing protein sequences of *Arabidopsis thaliana*, *Liriodendron chinese*, *Persea americana*, *Cinnamomum kanehirae*, and *Oryza sativa* in the UniProt and SwissProt databases (release-2020_05). For each reference, the following steps were implemented: (1) Putatively homologous genes were predicted from alignments with protein sequences representing the whole gene set (the longest transcripts for each gene were selected) with TBLASTN (2.2.18)[81] (e-value cut-off: 1e-5); and (2) the aligned regions were recovered together with sequences 2 kb downstream and upstream[82]. The alignments were further processed with GeneWise (2.2.0)[83] to generate exact exon and intron information. Prediction of evidence-based genes was carried out by aligning all the RNA-seq data with the assembled genome using Hisat2 (v2.0.4)[67], with cDNAs predicted by a genome-guided approach using StringTie (v1.2.2)[84] and then plotted against the genome using PASA (version 2.0.2);[85] the resulting cDNA sequence assembly (assembled using TRINITY; see below) was aligned to the *C. spicatus* genome sequences using BLAT (v.36)[65]. Using the MAKER pipeline (v2)[86] a non-redundant gene set was created that represented putative homologous genes, de novo genes, and RNA-seq supported genes, and for annotation, it was combined with a final set of 21,392 protein-coding genes.

*Chloranthus spicatus* was found to have a greater average gene length than most other angiosperms so far examined, so we decided that the complete start and stop codons should be used as the two boundaries for a gene and all the introns (exon-junction) must be supported by RNA-seq data (>2 reads). Determining gene length in this way produced the same result (Supplementary Data 8), which might be due to a long average intron length (Supplementary Data 9).

Two approaches were employed to evaluate the final gene set. First, the BUSCO core eukaryotic gene-mapping method was employed to check the completeness of the gene set, and second, RNA read mapping was used to evaluate the gene set. The RNA reads were mapped to the gene set using TopHat[87], whereas Samtools (v.0.1.19)[88] was used for the calculation of coverage depth. All data suggested complete and reliable gene annotation (Supplementary Data 4).

Functional annotation of the predicted genes was performed using a BLASTP homologue search against public protein databases, such as KEGG (59.3)[89], SwissProt (release-2020_05)[90], TrEMBL (release-2020_05), and the NCBI non-redundant protein NR database (20201015). InterProScan (v5.28–67.0)[91] was also used to provide functional annotation (Supplementary Data 22). KAAS (https://www.genome.jp/kegg/kaas/), KOBAS (http://kobas.cbi.pku.edu.cn/), and KOALA (https://www.genome.jp/tools/kofamkoala/) were utilized to explore the KEGG GENES database for KO (KEGG Orthology) assignments and to generate a KEGG pathway membership. We used the dbCAN2 metaserver (http://bcb.unl.edu/dbCAN2/index.php) for CAZyme annotation. For transcription factors, transcriptional regulators, and protein kinases, we used the online tool iTAK (v1.6)[92].

**Transcriptome assembly and gene expression analysis.** Before performing the transcriptome assembly, we retrieved high-quality reads by eliminating adaptor sequences and filtering low-quality reads using TRIMMOMATIC (v0.40)[53]. The resulting high-quality reads were then de novo assembled with TRINITY (v2.6.6)[66]. Protein sequences and coding sequences of transcripts were identified using TransDecoder (v5.3.0) (https://github.com/TransDecoder/TransDecoder), which also compares the translated coding sequences with the Pfam domain database[93]. Transcript abundance and coverage were calculated using the longest transcript for genes with multiple transcripts. The fragments per kilobase per million mapped reads (FPKM) approach was used to standardize transcript abundance levels, and FPKM values were calculated according to Mortazavi's method[94].

**Gene family identification.** The rate of protein evolution and the conservation of gene repertoires among orthologs in the genomes of 18 species were investigated using a comparative approach (Supplementary Data 7). We used BLASTP with an E-value of $1e^{-5}$ to align all-to-all proteins, and then OrthoMCL (1.4)[95] with a Markov inflation index of 1.5 and a maximum E-value of $1e^{-5}$ to cluster genes. All gene families from 18 reference genomes were acquired on this basis, and discovered genes belonging to *C. spicatus*-specific gene families and/or unclustered genes were detected.

This analysis yielded 34,552 gene families and 257 single-copy genes in *C. spicatus*, containing 19,614 predicted genes (91.7% of the total genes identified); orthologous genes in the 17 other species are displayed in Supplementary Data 7. To build a phylogenetic species tree, we employed low-copy families shared by all 18 species (see below). Using a random birth and death model, we used Café (v4)[96] to identify gene families that had expanded or contracted, and estimated the size of each family at each ancestral node, and provide a family-wise p-value (based on a Monte–Carlo resampling technique) to determine the presence or absence of expansion or contraction in each gene family.

**Phylogenetic analyses.** We constructed a phylogenetic tree based on three different conditions: 257 single-copy families in all 18 species (Supplementary Data 7) (1 gene for each species); 937 low-copy families at most missing three of 18 species (0 or 1 gene for each species, allowing a maximum of 3 species as missing); and 2,329 low-copy nuclear families (1, 2, or 3 gene copies for each species, with a maximum of 20 genes allowed) in all 18 species.

For each data set, we performed multiple amino acid sequence alignments using MAFFT (v.7.310)[97] for each single/low-copy gene orthogroup; DNA sequences were then aligned using PAL2NAL (v14)[98], followed by gap position removal using trimal (v1.4.1)[99] (only positions where 50% or more of the sequences have a gap were treated as a gap position). The maximum-likelihood (ML) software IQTREE (v 1.6.12)[100], coupled with ModelFinder[101], was used to reconstruct the phylogenetic tree for each single/low-copy orthogroup. The gene trees for each data set were then analyzed by ASTRAL (v.5.6.1)[102] to infer the species tree with quartet scores and posterior probabilities.

We built additional phylogenetic trees using extended species sampling as follows (Supplementary Data 7). One analysis was based on the concatenated matrix constructed from the 612 low-copy orthologs clustered by OrthoFinder (v2.2.7)[103] from 218 species (215 published genomes + two transcriptomes (1KP), and the newly generated *C. spicatus* genome in this study) using IQTREE (v 1.6.12)[100]. A second analysis was based on 80 gene sequences representing rRNA and protein-coding genes from 134 published plastid genomes, analyzed with RAxML (v. 8.2.12)[104] using GTRGAMMA and GTRCAT modes separately with 100 bootstrap replicates.

**Divergence time estimation.** To estimate divergence times, we used MCMCTREE (v4.5) of the PAML package[105]. The Markov chain Monte Carlo (MCMC) procedure was performed for 1,500,000 iterations at 150 sample frequency after a burn-in of 500,000 iterations The default settings of MCMCTREE were utilized for all other parameters. Convergence was checked by two independent runs. The data set with the highest number of genes (2,329 LCN gene families) was used as the input file for MCMCTree, and multiple carefully vetted fossils (Supplementary Data 11b) were used as calibrations following[21] and Timetree (http://www.timetree.org/).

**Analysis of genome synteny and whole-genome duplication.** To estimate the timing of whole-genome duplication events, gene families based on OrthoMCL (v 1.4) analysis were used[95], followed by manual filtering to exclude low-copy families based on both pairwise comparison of paralog sequences (within the species genome, $1 < n < 5$) and orthologous relationships (between *C. spicatus* and other species, 1:1).

The alignment of the gene families was performed with MUSCLE (v3.8.31)[106], and using the CODEML program of the PAML package[105]. The *Ks* estimates for all the pairwise comparisons within a gene family were generated by maximum likelihood estimation. To correct the redundancy of *Ks* values (a gene family of n members produces $n*(n-1)/2$ pairwise *Ks* estimates for $n-1$ retained duplication events), a phylogenetic tree was built for each subfamily with IQTREE (v 1.6.12)[100] following default settings. All m *Ks* estimates between the two child (sub) clades were added to the *Ks* distribution with a weight of $1/m$ (where *m* is the number of *Ks* estimates for a duplication event), so that the weights of all *Ks* estimates for a single duplication event summed to one for each duplication node in the resulting phylogenetic tree.

The age of the WGD event found here for *C. spicatus* was estimated by combining the *Ks* value with synonymous substitutions at each site per year (*r*) by using equation 1 (see above in LTR insertion time estimation). The timing of this event was also compared to the evidence for WGD in other magnoliids (Supplementary Data 11a, b).

Genome-wide syntenic and collinear blocks were identified both within and among selected genomes. Based on the result with all vs. all Blastp (E-value ≤ $1e^{-5}$), the investigation was carried out on the translated protein sequences of the 18 annotated gene models, establishing a database of protein similarity. Then the Multiple Collinearity Scan (MCScanX, v0.8, >5 homologous gene pairs/block)[107] was used to identify conserved collinear blocks.

**Distinguishing hybridization from ILS.** We used QuIBL, a tree-based method of assessing branch length distribution across gene trees, to quantify putative hybridization[35]. First, we ran QuIBL using default settings. The 100 runs of QuIBL were conducted with 500 trees selected randomly from the 2,329 gene trees resulting from the LCN gene families, as described above, for each run. For the results of QuIBL, we formally distinguished between an ILS-only model and a hybridization model using the Bayesian Information Criterion (BIC) test with a strict cutoff of $\Delta BIC > 10$. Then, we calculated the average ILS/hybridization ratio from 100 runs and defined the topology structure type (T1, T2, T3) according to the number of gene trees supporting each topology. Finally, we calculated the average ratio for each pair from all triplets' results, for example, calculating the real outgroup A & C with topology T1 (A,(Others, C)) and considering the remaining species as Others. Discordant triplets (Others,(A, C)) were then used to calculate the average ILS/ hybridization ratio between pair A&C.

Using a maximum pseudo-likelihood technique, we inferred species networks that model ILS and gene flow[108]. Analyses were carried out with PhyloNet v.3.8.2[109] with the command 'InferNetwork_MPL' using the 2,329 LCN gene trees as input. Five parallel network searches were carried out by allowing one to five reticulation events. To estimate the best network from each of the five maximum reticulation events searches, we performed model selection using the bias-corrected Akaike Information Criterion (AICc)[110] and the Bayesian Information Criterion (BIC)[111]. The number of parameters was set to the number of branch lengths being estimated plus the number of hybridization probabilities being estimated[112]. To test the potential for false positives, we also generated coalescent trees under the species tree with ILS as the only underlying source of conflict. The ASTRAL tree of 18 species was used as input, and 2,329 trees (matching the number of empirical gene trees) were simulated in DendroPy[113] in 10 replicates. A total of 2,329 randomly selected simulated trees from the resulting pool of 23,290 trees was then used to repeat the PhyloNet analyses.

To assess whether the observed phylogenetic incongruences are mainly due to hybridization, we conducted the ABBA-BABA D-statistic test[114,115]. The D-statistic considers a topology (((P1, P2), P3), O) and can detect the existence of hybridization between P1 and P3 or P2 and P3. We limited our taxon sampling to one outgroup and seven ingroup taxa to cover representative species from each clade due to computational constraints, and given our main focus on identifying potential reticulating events among magnoliids, eudicots, and monocots. Specifically, genomic data of seven species of mesangiosperms, plus *Nymphaea colorata* (Nymphaeales), which was used as the outgroup, were downloaded from NCBI (*Arabidopsis thaliana*: SRR13376045, *Vitis vinifera*: SRR12328991, *Liriodendron chinense*: SRR6876701, *Oryza sativa*: ERR3087460, *Musa acuminata*: ERR3412984, *Phalaenopsis equestris*: SRR827630, *Nymphaea colorata*: SRR10158655). Then, BAM files of all 8 species were used as an input for D-statistics investigation. The program doAbbababa2 (http://www.popgen.dk/angsd/index.php/Abbababa2) implemented in ANGSD[116] was used to calculate the D-statistics for all the species or groups, and *N. colorata* was used as the outgroup in all four-taxon groups. The results from the doAbbababa2 program included the D-statistics for all potential four-taxon combinations of species or clades[117]. Only the results in which the topology conformed to the phylogeny (based on the 18-species nuclear data set, Fig. 2a) were considered. The absolute value of a Z score greater than 3 was considered to be significant, and gray lines are used to mark the non-significant results in Supplementary Fig. 19.

**Analysis of TPS genes and the terpene synthesis pathway**. To identify the genes involved in terpenoid synthesis, we referred to the TERPENOID BACKBONE BIOSYNTHESIS (Map00900) in KEGG (https://www.genome.jp/kegg) with relevant genes from *A. thaliana* as bait[89]. The corresponding genes in *C. spicatus* were obtained by combining genome annotation using KOfamKOALA and blastp[62] results (E-value<10$^{-5}$, identity ≥50%, and coverage ≥30%). Gene expression was measured by log$_2$ (FPKM) and visualized by the pheatmap package (v1.0.12) in R. The TPS duplicated genes were identified using DupGen_finder[118], and were classified into whole-genome duplication (WGD) and tandem duplicates (TD). To identify the most recent and/or ancient *Ks* peaks (or WGDs) for TPS genes, the *Ks* values for the gene pairs in each syntenic block were estimated. The *Ks* values in the range of about 1.0 +/- 0.14 were used to identify TPS duplicated genes.

Candidate TPSs were identified from proteomes and transcriptomes to minimize missing potential TPS genes. First, the published TPS genes of *A. thaliana*, *O. sativa*, *C. kanehirae*, and *L. cubeba* and the TPSs of Embryophyta published in NCBI (search keywords: Embryophyta [Organisms] AND terpene synthase) were used as reference sequences to align *C. spicatus* sequence by blastp. Second, to search the proteomes by HMMER (v3.2.1)[119], two Pfam domains (PF01397 and PF03936) were used. Third, the transcriptome data were used to check for missing TPSs due to genome sequencing or assembly errors. Finally, 73 TPS genes in *C. spicatus* were obtained by integrating the above results and eliminating redundancy with a consistent standard (E-value<10$^{-5}$, ≥200 amino acids). Maximum likelihood trees of TPS genes were built using IQTREE (v1.6.12)[100] after alignment in MAFFT (v7.310);[97] subfamily classification of sequence types was based on *A. thaliana* TPSs. The same method was used to identify candidate TPS genes in the remaining 13 of the 18 angiosperms analyzed.

For phylogenetic analysis of the cytochrome P450 gene family, the protein sequences of other P450 sequences were also retrieved from the uniport database (Q8W1W8: CYP719A *Coptis japonica*, Q8W1W8: CYP726A1 *Euphorbia lagascae*, C99A1_SORBI: CYP99A1 *Sorghum_bicolor*, C99A2_ORYSJ: CYP99A2 *Oryza sativa*, C7A12_PANGI: CYP736A12 *Panax ginseng*, C92C5_MAIZE: CYP92C5 *Zea mays*, C92C6_MAIZE: CYP92C6 *Zea mays*, C80A1_BERST: CYP80A1 *Berberis stolonifera*, C80B1_ESCCA: CYP80B1 *Eschscholzia californica*, C80B3_PAPSO: CYP80B3 *Papaver somniferum*). Then MAFFT (parameters:–anysymbol–maxiterate 1000 –localpair) was used for the alignment followed by the construction of a phylogenetic tree using IQTREE (v1.6.12)[100] (parameters: -bb 5000 -alrt 1000).

**Characterization of the *R* genes**. NBS-LRR (NBS for short) genes encompass more than 80% of the characterized *R* genes[40]. The NBS genes are classified into three classes, namely, TIR-NBS-LRR (TNL), CC-NBS-LRR (CNL), and RPW8-NBS-LRR(RNL)[40]. In all, 3,518 NBS genes were identified in 28 angiosperm species, including 10 that were previously reported[40] and the 18 species of angiosperms analyzed in this study (Supplementary Data 7, Fig. 4c). The protein sequences of the 18 newly analyzed species were matched to the LRR_1 (PF00560.27), LRR_2 (PF07723.7), LRR_3 (PF07725.6), LRR_4 (PF12799.1), NB-ARC (PF00931.16), Pkinase (PF00069.19), RPW8 (PF05659.10), TIR (PF01582.14), and zf-BED (PF02892.10) domains using the HMMER search software with an E-value cut-off of 1e$^{-05}$. The Coiled-Coil (CC) domain was predicted using paircoil2 (http://cb.csail.mit.edu/cb/paircoil2/) with a threshold score of 0.025 for filtering the Coiled-Coil (CC) domain search results.

**Reporting summary**. Further information on research design is available in the Nature Research Reporting Summary linked to this article.

## Data availability
The data supporting the findings of this work are available within the paper and its Supplementary Information files. A reporting summary for this article is available as a Supplementary Information file. The genome and transcriptome sequence data generated in this study have been deposited in the NCBI Sequence Read Archive database under accession code PRJNA770110, CNSA of CNGBdb with accession code CNP0001771, and the assembled genome and annotation have been deposited in the Genome Warehouse in National Genomics Data Center, Beijing Institute of Genomics, Chinese Academy of Sciences/China National Center for Bioinformation under accession number GWHBFSJ00000000. Source data are provided with this paper.

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

## Acknowledgements

This research was supported by grants from the National Natural Science Foundation of China (Grant No. 32000171) awarded to X. Guo. This work was also supported by the National Key R&D Program of China (No. 2019YFC1711000), Major Science and Technology Projects of Yunnan Province (Digitalization, development and applica-tion of biotic resource (No. 202002AA100007), Shenzhen Municipal Government of China (No. JCYJ20170817145512476) and Guangdong Provincial Key Laboratory of Genome Read and Write (No. 2017B030301011). This work is part of the 10KP project (https://db.cngb.org/10kp/). This work is also supported by China National GeneBank (CNGB; https://www.cngb.org/). We are grateful to Zhongjian Liu, Matt Gitzendanner, Fang Wang, Yinghui Wang, Lin Xian, Hui Zhang, Jin Pan, Yewen Chen, Weixue Mu, Yannan Fan, and Yalong Guo for general technical assistance or discussion; Shuqiang Chen for collecting the plant materials; and Qiang Zhang for providing photographs.

## Author contributions

H.L. and D.S. led and designed this project. X.Guo and D.F. conceived the study. X.G., D.F., and S.K.S. wrote the manuscript. D.F. and S.Y. generated the whole-genome assembly. D.F., X.Guo., S.Y., X.G., S.C., M.L., T.Y., and S.K.S. performed the functional annotation, comparative genomics, and transcriptome data analyses, and generated the figures. H.L., D.S., P.S., R.F., S.Z., S.S., X.L., X.X., and A.S.C. revised and edited the manuscript. All authors read and approved the final paper.

## Competing interests

The authors declare no competing interests.
