## [Peer Review File · Nature Communications]

Chloranthus genome provides insights into the early diversification of angiospermsReviewers' Comments:

Reviewer #1:

Remarks to the Author:

NCOMMS-21-19738

Lead authors Guo, Fanng, Sahu and the international team of co-workers report the genome of a *Chloranthus* species. Plants of this clade are known for their use in traditional medicine due to accumulation of a rich repertoire of bioactive phytochemicals (terpenoids) and for their ambiguous phylogenetic position within the early radiating angiosperms.

This study predominantly focuses on the latter and describes sequencing and high-quality chromosome scale assembly and annotation of the nuclear genome of *C. spicatus*, which revealed intriguing characteristics, next to solving the phylogenetic relationship of the Chloranthales within the angiosperms. Despite carrying a total number of genes in the same range as other plant lineages, the genome was found expanded in size and rich in repetitive transposable elements (TEs). These apparently contribute to an increase in overall gene (intron) size with a possible impact on gene expression.

The team reports an ancient whole-genome duplication, specific for the Chloranthales, but predating their speciation. While a WGD is not known to have occurred in the phylogenetically more basal conifer lineage, their genomes also underwent expansion with accumulation of large numbers of TEs, indicating a dynamic role in the evolution of these genomes. In support, the authors highlight transposition as mechanism to contribute to a third of the duplication events specifically of the terpene synthase (TPS) gene family. Following duplication, TPS of subfamilies TPS-a and TPS-b were retained in the genome at a notable rate. These subfamilies contain in angiosperms typically, but not exclusively, TPS involved in formation of sesqui- and monoterpenes, respectively, and with that could contribute to the rich phytochemistry in the Chloranthales.

The manuscript is written very accessible for a genomics non-specialist. The argumentation is clear, and the data is presented concise. The study solves a long-standing question of the evolution of a major angiosperm lineage and highlights genomic features characteristic of this clade. While not within the scope of this study, the genomic resources developed here clearly prompt future functional studies. Compared to the earlier reported genomes of conifers the paper is in my opinion of interest for a broad audience and represents a major advance in the field.

Minor comments and questions, in order of the manuscript

- L123: Is the high heterozygosity rate reported connected to reproduction of *C. spicatus*, i.e. is this plant species an obligate outbreeder?
- LTR elements: In mono- and dicotyledon lineages, up to several million copies of LTR retrotransposons constituting 80% of the contemporary genomes were suggested to have emerged through bursts of activity over the last 3-11 MY. With few exceptions, in gymnosperms that number dates much further back. You describe the relative comparison of the timing between the LTRs in genes and those found distributed in the genome. If available, using the molecular clock/average synonymous substitution rate (μS), the timing of insertion of LTRs could be dated for this plant lineage.
- You describe the 'ALL LTR' as distributed anywhere in the genome. In other angiosperms, the genomic distribution of LTR elements was not found homogenous. Could you give a map of the distribution of LTRs over the genome?
- TPS: It may provide relevant context to introduce the unusual structures of terpenes found in *Chloranthus*. They have shown to be bioactive and were target of several strategies for formal chemical synthesis.
- Are any of the TPSs found in gene clusters with P450s, typically involved with further functionalization of the backbone? These may also, in conjunction with additional decoration result in an increase in polarity and loss of volatility.
- L384 Wording; The MEP and MEV pathways are not involved in 'primary volatile substances', but an

entire spectrum of small to much larger, non-volatile isoprenoids.

- L390 Wording; The 'HMG protein' should be Gene encoding the HMGR enzyme, or HMG CoA-reductase.

- L392, and throughout: GGDPS, or GGPPS, geranylgeranyl diphosphate synthase

- L402 Wording; 'which is responsible for the production of sesquiterpenes', should be written more carefully, unless functionally supported. Note: there are reports of TPS other than sesqui- and monoTPS in both TPS-a and TPS-b.

- The references should be carefully proof-read for consistency.

- Fig. 4a, Comments: It is unclear what the relevance of the cis-prenyl intermediate NPP is in this species. Suggestion: remove, or support by reported products in that configuration. The figure focuses on specialized metabolites, i.e., no sterols or carotenoids. I suggest removing the cytosolic GGDPS and its product.

Bjoern Hamberger

Reviewer #2:

Remarks to the Author:

This is a well written manuscript with high-quality figures that deals with a competently performed genome sequencing of plant whose lineage has previously not been sequenced at whole-genome level.

What are the noteworthy results? First, in the authors' own words, this work provides a "valuable genomic resource for future investigations". The main result, and the motivation for the study, is improved resolution of the phylogenetic position of this lineage. That position is controversial with nuclear loci presenting a different tree topology compared with nuclear loci.

Will the work be of significance to the field and related fields? Yes, as the authors say, this provides a valuable resource for future work, providing an important missing piece in the genomic coverage of plant diversity. Further, the phylogenetic position is now resolved as far as it is ever likely to be (as a result of this work). What is not entirely clear to me is how much of an advance is this phylogenetic knowledge compared with what we already knew based on a few previously sequenced loci. I would have liked to see the authors explicitly stating what we now know that we did not know already before sequencing this nuclear genome.

How does it compare to the established literature? If the work is not original, please provide relevant references. The work is original. Compared with other descriptions of the genome sequencing of a plant it is well written, with good documentation of the methods used and a good overview description of the genome sequence.

Does the work support the conclusions and claims, or is additional evidence needed? Yes, the evidence backs the authors' conclusions to the extent that they present them. In other words, the authors are honest about the remaining uncertainties. But I am still not clear about whether the phylogeny based on nuclear genome-wide data is significantly different from that based on a few nuclear loci. The authors also mention that observed incongruities in the trees could be explained by incomplete lineage sorting and/or hybridization and make an attempt to distinguish these. I was left slightly unsure what was the outcome of that attempt. Do we now know? I note that the authors did identify a 'new' previously unknown and distinct genome duplication that adds to our knowledge.

Are there any flaws in the data analysis, interpretation and conclusions? Do these prohibit publication

or require revision?

Is the methodology sound? Does the work meet the expected standards in your field? Is there enough detail provided in the methods for the work to be reproduced? Yes, the paper is well written and complete in these respects.

A few very minor points:

Sometimes in the figures the authors divide the Cretaceous into 'Upper' and 'Lower' but elsewhere they talk about 'Early Cretaceous'. This inconsistency is potentially confusing.

I am not sure what is the rationale for using italics for 'Mesangiospermae'.

Around line 124: Does Gb here mean gigabytes or gigabases? I presume the latter.

Line 215. Is the definition of Ks correct here?

Line 263: "poor taxon sampling may lead to topological errors, we added ... to increase our taxon sampling". So, by 'poor' do the authors really mean 'sparse'?

Lines 365-367. This section is slightly confusing. The authors mention 'GO analyses' but this phrase tells us nothing about what kind of analysis this is. On examining Table S17 it becomes apparent that this analysis is the identification of enrichment of GO terms. The authors then mention large P values; so does this refer to statistical non-significance? Presumably the most enriched will have low P values? Please consider making this section more explicitly clear.

Line 405 "Chloranthus has a subset of eudicot R genes": This statement is without value. Of course it has a subset of the R genes. Even if it contained all of the eudicot R genes, that would still be a subset, albeit a big one! And if it had no R genes then the statement would still be true as the empty set is a subset of all sets.

Line 424. The name of the company/brand is not 'Nanopore'; it is Oxford Nanopore Technologies. The generic name of the technology is not 'Nanopore'; it could be 'nanopore'.

Line 507. Is there a reference that could be cited for MinKnow?

Lines 541 to Line 543. How were these parameter values optimised? To what extent was the assembly more- or less-accurate and/or contiguous when different parameter values were chosen?

When discovering repetitive elements, how did the authors distinguish between 'repeats' and families of paralogous genes?

TopHat2 not Tophat2.

Pfam not PFAM.

In the PDF of Figure 1, something has gone slightly wrong with the 'Ma', as if the image has been slightly truncated. Please check whether anything is missing from the figure.

Reviewer #3:

Remarks to the Author:

The authors present the first genome of Chloranthales and resolve important outstanding questions on

the diversification of angiosperms that were difficult to address prior to the new data presented. The combination of the *Chloranthus* genome, robust comparative analyses testing for the contributions of incomplete lineage sorting and hybridization, and patterns of diversification of important gene families make compelling contributions to our understanding of angiosperm diversification. I do not have any substantive changes to suggest.

RESPONSE TO REVIEWER COMMENTS

Reviewer #1 (Remarks to the Author):

NCOMMS-21-19738

Lead authors Guo, Fang, Sahu and the international team of co-workers report the genome of a *Chloranthus* species. Plants of this clade are known for their use in traditional medicine due to accumulation of a rich repertoire of bioactive phytochemicals (terpenoids) and for their ambiguous phylogenetic position within the early radiating angiosperms.

This study predominantly focuses on the latter and describes sequencing and high-quality chromosome scale assembly and annotation of the nuclear genome of *C. spicatus*, which revealed intriguing characteristics, next to solving the phylogenetic relationship of the Chloranthales within the angiosperms. Despite carrying a total number of genes in the same range as other plant lineages, the genome was found expanded in size and rich in repetitive transposable elements (TEs). These apparently contribute to an increase in overall gene (intron) size with a possible impact on gene expression.

The team reports an ancient whole-genome duplication, specific for the Chloranthales, but predating their speciation. While a WGD is not known to have occurred in the phylogenetically more basal conifer lineage, their genomes also underwent expansion with accumulation of large numbers of TEs, indicating a dynamic role in the evolution of these genomes. In support, the authors highlight transposition as mechanism to contribute to a third of the duplication events specifically of the terpene synthase (TPS) gene family. Following duplication, TPS of subfamilies TPS-a and TPS-b were retained in the genome at a notable rate. These subfamilies contain in angiosperms typically, but not exclusively, TPS involved in formation of sesqui- and monoterpenes, respectively, and with that could contribute to the rich phytochemistry in the Chloranthales.

The manuscript is written very accessible for a genomics non-specialist. The argumentation is clear, and the data is presented concise. The study solves a long-standing question of the evolution of a major angiosperm lineage and highlights genomic features characteristic of this clade. While not within the scope of this study, the genomic resources developed here clearly

prompt future functional studies. Compared to the earlier reported genomes of conifers the paper is in my opinion of interest for a broad audience and represents a major advance in the field.

Minor comments and questions, in order of the manuscript

- L123: Is the high heterozygosity rate reported connected to reproduction of *C. spicatus*, i.e. is this plant species an obligate outbreeder?

Response: Yes, *C. spicatus* is an obligate outbreeder. The revised sentence now reads as follows “the individual sequenced had a heterozygosity rate of 0.99%, which is possibly associated with the obligate outcrossing system of this species. Line 116-118.

von Balthazar, M., & Endress, P. K. (1999). Floral bract function, flowering process and breeding systems of *Sarcandra* and *Chloranthus* (Chloranthaceae). *Plant Systematics and Evolution*, 218(3), 161-178.

- LTR elements: In mono- and dicotyledon lineages, up to several million copies of LTR retrotransposons constituting 80% of the contemporary genomes were suggested to have emerged through bursts of activity over the last 3-11 MY. With few exceptions, in gymnosperms that number dates much further back. You describe the relative comparison of the timing between the LTRs in genes and those found distributed in the genome. If available, using the molecular clock/average synonymous substitution rate (μS), the timing of insertion of LTRs could be dated for this plant lineage.

Response: Added. The description reads as follows “The time of insertion of LTRs in the genome of *C. spicatus* was estimated based on a peak substitution rate at 0.03 (Supplementary Fig. 4). We assumed a synonymous substitution rate of 1.51×10^{-9} bases per year following two recent studies of the closely related magnoliid lineages *Liriodendron*¹⁶ and *Chimonanthus*¹⁷, resulting in an LTR burst time of approximately 9.9 Ma.” Line 139-143

Supplementary Fig. 4 Estimated distribution of full-LTR Insertion Times in *C. spicatus* genome. Ks distributions of the full-LTR in the *C. spicatus* genome was plotted by a window of 0.005, and a Ks peak was found at 0.03. We assumed a mutation rate of 1.51×10^{-9} bases per year (Chen et al 2019), resulting in an insertion time of approximately 9.9 Ma.

Chen JH, Hao ZD, Guang XM, Zhao CX, Wang PK, Xue LJ, et al. Liriodendron genome sheds light on angiosperm phylogeny and species–pair differentiation. Nat Plants. 2019;5:18.

- You describe the ‘ALL LTR’ as distributed anywhere in the genome. In other angiosperms, the genomic distribution of LTR elements was not found homogenous. Could you give a map of the distribution of LTRs over the genome?

Response: We used ‘(anywhere in the genome)’ to define ‘ALL LTR’. A map was added to show the distribution of LTRs in 15 chromosomes. We agree that the distribution of LTR elements was not homogenous, and found to be slightly more abundant in centromeric regions. To avoid ambiguity, the sentence was revised as follows “We discovered 11,500 intact LTRs and classified them into two groups, Gene-20K LTR (LTRs distributed in genes >20 kb length) and ALL LTR (LTRs distributed throughout the genome, Supplementary Fig. 6).” Line 181-183

Supplementary Fig. 6 Density distribution of the full-LTR in *C. spicatus* genome. All numbers were determined in 2-Mb windows.

- TPS: It may provide relevant context to introduce the unusual structures of terpenes found in *Chloranthus*. They have shown to be bioactive and were target of several strategies for formal chemical synthesis.

Response: Several unique *Chloranthus*-specific sesquiterpenoids, including Chlorahololides A, Chloranthalactone A, Chlotrichenes A and B with bioactive potential, have been isolated and chemically synthesized in the lab^{37, 38, 39}. Line 381-384.

Liu, Y., & Nan, F. J. (2010). Synthetic studies towards Chlorahololides A: practical synthesis of a lindenane-type sesquiterpenoid core framework with a 5, 6-double bond. Tetrahedron Letters, 51(10), 1374-1376.

Yue, G., Yang, L., Yuan, C., Jiang, X., & Liu, B. (2011). Total synthesis of (±)-chloranthalactone A. Organic letters, 13(19), 5406-5408.

*Chi, J., Xu, W., Wei, S., Wang, X., Li, J., Gao, H., ... & Luo, J. (2019). Chlotrichenes A and B, two lindenane sesquiterpene dimers with highly fused carbon skeletons from *Chloranthus holostegius*. Organic letters, 21(3), 789-792.*

- Are any of the TPSs found in gene clusters with P450s, typically involved with further functionalization of the backbone? These may also, in conjunction with additional decoration result in an increase in polarity and loss of volatility.

Response: Thanks for raising this interesting point. According to your suggestion, we performed a phylogenetic analysis of cytochromes P450 super gene families between *Amborella trichopoda*, *Liriodendron chinense*, *Chloranthus spicatus*, *Arabidopsis thaliana*, and *Oryza sativa*, and We found considerable expansion of P450 gene families such as CYP71B, CYP84, CYP706, CYP78, CYP79A, CYP72, CYP719 in *C. spicatus*, suggesting their plausible role in functionalization of the TPSs backbone⁴⁷ (Supplementary Fig. 23). Based on previous studies, most terpenoid-related cytochromes P450 are members of the CYP71 clade, and are involved in the formation of the sesquiterpene ester ring^{37, 48}. We also discovered a relatively high number of copies in the CYP71 gene family in *C. spicatus* in comparison to its magnoliid relative, *L. chinense* (Supplementary Fig. 23)., which provided further cues regarding the higher volatile content in *C. spicatus*. The relevant description has been added in the discussion part as well. Line 501-510

De Kraker, J. W., Franssen, M. C., Joerink, M., De Groot, A., & Bouwmeester, H. J. (2002). Biosynthesis of costunolide, dihydrocostunolide, and leucodin. Demonstration of cytochrome P450-catalyzed formation of the lactone ring present in sesquiterpene lactones of chicory. Plant physiology, 129(1), 257-268.

Hamberger, B., & Bak, S. (2013). Plant P450s as versatile drivers for evolution of species-specific chemical diversity. Philosophical Transactions of the Royal Society B: Biological Sciences, 368(1612), 20120426.

Banerjee, A., & Hamberger, B. (2018). P450s controlling metabolic bifurcations in plant terpene specialized metabolism. Phytochemistry Reviews, 17(1), 81-111.

Supplementary Fig. 23 The phylogenetic tree of cytochromes P450 gene family

Each color represents individual species, and the table shows the gene copy number of each

expanded gene families in *C. spicatus*. pfam (PF00067) was used to search the protein sequences of the five species with E-value cut off of 1e-05.

Methodology (Line 847-856): For phylogenetic analysis, the protein sequences of other P450 sequences were also retrieved from the uniprot database (Q8W1W8: CYP719A *Coptis japonica*, Q8W1W8: CYP726A1 *Euphorbia lagascae*, C99A1_SORBI: CYP99A1 *Sorghum bicolor*, C99A2_ORYSJ: CYP99A2 *Oryza sativa*, C7A12_PANGI: CYP736A12 *Panax ginseng*, C92C5_MAIZE: CYP92C5 *Zea mays*, C92C6_MAIZE: CYP92C6 *Zea mays*, C80A1_BERST: CYP80A1 *Berberis stolonifera*, C80B1_ESCCA: CYP80B1 *Eschscholzia californica*, C80B3_PAPSO: CYP80B3 *Papaver somniferum*). Then the MAFFT (parameters: --anysymbol --maxiterate 1000 --localpair) was used for the alignment followed by construction of phylogenetic tree using IQTREE (v1.6.12)¹⁰¹ (parameters: -bb 5000 -alrt 1000).

- L384 Wording; The MEP and MEV pathways are not involved in ‘primary volatile substances’, but an entire spectrum of small to much larger, non-volatile isoprenoids.

Response: The sentence has been revised as “Next, we investigated the genes involved in the production of non-volatile isoprenoids via the 2-C-methyl-D-erythritol 4-phosphate (MEP) pathway and the mevalonate (MVA) pathway”

- L390 Wording; The ‘HMG protein’ should be Gene encoding the HMGR enzyme, or HMG CoA-reductase.

Response: The sentence has been revised as “The gene encoding the HMGR enzyme (Hydroxy-3-methylglutaryl) displayed the highest number of gene copies...”

- L392, and throughout: GGDPS, or GGPPS, geranylgeranyl diphosphate synthase

Response: Here GGPS stands for geranylgeranyl pyrophosphate synthase. We have corrected the typing mistake.

- L402 Wording; ‘which is responsible for the production of sesquiterpenes’, should be written

more carefully, unless functionally supported. Note: there are reports of TPS other than sesqui- and monoTPS in both TPS-a and TPS-b.

Response: We toned down the statement as “which is **probably** responsible for the production of sesquiterpenes in Chloranthales”

- The references should be carefully proof-read for consistency.

Response: We carefully proofread all the references.

- Fig. 4a, Comments: It is unclear what the relevance of the cis-prenyl intermediate NPP is in this species. Suggestion: remove, or support by reported products in that configuration. The figure focuses on specialized metabolites, i.e., no sterols or carotenoids. I suggest removing the cytosolic GGDPs and its product.

Response: Thanks for the nice suggestion. We have removed the cytosolic GGDPs and its product from Fig. 4a (left panel)

Reviewer #2 (Remarks to the Author):

This is a well written manuscript with high-quality figures that deals with a competently performed genome sequencing of plant whose lineage has previously not been sequenced at whole-genome level.

What are the noteworthy results? First, in the authors' own words, this work provides a "valuable genomic resource for future investigations". The main result, and the motivation for the study, is improved resolution of the phylogenetic position of this lineage. That position is controversial with nuclear loci presenting a different tree topology compared with nuclear loci.

Will the work be of significance to the field and related fields? Yes, as the authors say, this provides a valuable resource for future work, providing an important missing piece in the genomic coverage of plant diversity. Further, the phylogenetic position is now resolved as far as it is ever likely to be (as a result of this work). What is not entirely clear to me is how much of an advance is this phylogenetic knowledge compared with what we already knew based on a few

previously sequenced loci. I would have liked to see the authors explicitly stating what we now know that we did not know already before sequencing this nuclear genome.

Response: The details are available in introduction section lines 60-99.

How does it compare to the established literature? If the work is not original, please provide relevant references. The work is original. Compared with other descriptions of the genome sequencing of a plant it is well written, with good documentation of the methods used and a good overview description of the genome sequence.

Does the work support the conclusions and claims, or is additional evidence needed? Yes, the evidence backs the authors' conclusions to the extent that they present them. In other words, the authors are honest about the remaining uncertainties. But I am still not clear about whether the phylogeny based on nuclear genome-wide data is significantly different from that based on a few nuclear loci. The authors also mention that observed incongruities in the trees could be explained by incomplete lineage sorting and/or hybridization and make an attempt to distinguish these. I was left slightly unsure what was the outcome of that attempt. Do we now know? I note that the authors did identify a 'new' previously unknown and distinct genome duplication that adds to our knowledge.

Response: To summarize, the observed gene tree incongruence between nuclear and chloroplast trees and among nuclear single-gene trees suggested the possibility of ILS and/or hybridization events during early angiosperm evolution. Therefore, we performed a detailed analyses to confirm that ancient hybridization may account for the incongruent phylogenetic placement of Chloranthales + magnoliids relative to monocots and eudicots in nuclear and chloroplast trees

Given the lack of clarity felt by this reviewer as regards the ultimate outcome of the hybridization analyses we implemented, we added a rewritten conclusions paragraph to the hybridization section of the Results (Lines 351 to 357) as “*In summary, all three methods to investigate hybridization (QuIBL, PhyloNet, and ABBA-BABA D-statistics) were unanimous in suggesting ancient gene flow between monocots and eudicots, although with variation among methods in the number of hybridization events and any further lineages involved in*

hybridization. A consensus scenario is presented (Fig. 2c) showing ancient gene flow between monocots and eudicots.”

We also added a note on broader implications of this result in the Discussion (lines 493 to 500) as “*Our study suggests ancient gene flow is more likely than incomplete lineage sorting as the cause of the incongruent phylogenetic placement of magnoliids. The key role of ancient hybridization revealed here sheds additional light on the difficulty of inferring the branching order of major angiosperm lineages, traditionally explained by short divergence times alone. Our findings also raise important questions about the relative roles of different diversification mechanisms for the early explosion of angiosperms..”*

Are there any flaws in the data analysis, interpretation and conclusions? Do these prohibit publication or require revision?

Is the methodology sound? Does the work meet the expected standards in your field? Is there enough detail provided in the methods for the work to be reproduced? Yes, the paper is well written and complete in these respects.

A few very minor points:

Sometimes in the figures the authors divide the Cretaceous into 'Upper' and 'Lower' but elsewhere they talk about 'Early Cretaceous'. This inconsistency is potentially confusing.

Response: To make it consistent with the figure, 'Early Cretaceous' has been changed to “lower” Cretaceous'

I am not sure what is the rationale for using italics for 'Mesangiospermae'.

Response: '*Mesangiospermae*' is a published phylocode name for the clade as such, these are written in Italics based on the previous literature.

Soltis, P. S., & Soltis, D. E. (2016). Ancient WGD events as drivers of key innovations in angiosperms. *Current opinion in plant biology*, 30, 159-165.

Cantino, P. D., Doyle, J. A., Graham, S. W., Judd, W. S., Olmstead, R. G., Soltis, D. E., ... & Donoghue, M. J. (2007). Towards a phylogenetic nomenclature of Tracheophyta. *Taxon*, 56(3), E1-E44.

Around line 124: Does Gb here mean gigabytes or gigabases? I presume the latter

Response: Yes, GB stands for gigabases, and has been mentioned in the main text as well now.

Line 215. Is the definition of Ks correct here?

Response: We have revised the sentence as “*The distribution of Ks values, the number of synonymous substitutions per synonymous site, for*”

Line 263: "poor taxon sampling may lead to topological errors, we added ... to increase our taxon sampling". So, by 'poor' do the authors really mean 'sparse'?

Response: Yes, it means “sparse”. We have revised the sentence to avoid the confusion.

“*As sparse taxon sampling may lead to topological errors....*”

Lines 365-367. This section is slightly confusing. The authors mention 'GO analyses' but this phrase tells us nothing about what kind of analysis this is. On examining Table S17 it becomes apparent that this analysis is the identification of enrichment of GO terms. The authors then mention large P values; so does this refer to statistical non-significance? Presumably the most enriched will have low P values? Please consider making this section more explicitly clear.

Response: Thanks for pointing out the typing mistake. Indeed, the most enriched will have a low P value, and according Table S17, terpene synthase activity has the lowest p value. We also revised the sentence in the main text for better clarity as “Furthermore, when we performed GO enrichment analyses using the shared genes between magnoliids and Chloranthales, the genes related to terpene synthase activity (GO:0010333) exhibited a low P-value, indicating the terpene synthase activity was the most enriched among all GO categories (Supplementary Table S17)”

Line 405 "Chloranthus has a subset of eudicot R genes": This statement is without value. Of

course it has a subset of the R genes. Even if it contained all of the eudicot R genes, that would still be a subset, albeit a big one! And if it had no R genes then the statement would still be true as the empty set is a subset of all sets.

Response: The caption has been revised as “Distribution of R genes (disease resistance gene family) in *Chloranthus*”

Line 424. The name of the company/brand is not 'Nanopore'; it is Oxford Nanopore Technologies. The generic name of the technology is not 'Nanopore'; it could be 'nanopore'.

Response: The generic name has been revised as “nanopore”, while “Oxford Nanopore Technologies” is already used in the method and results section.

Line 507. Is there a reference that could be cited for MinKnow?

Response: MinKNOW refers to Oxford Nanopore Technologies Device Control software that is embedded in the MinIT, GridION, and PromethION; it is provided for installation on PCs for control of MinIONs. There is no published paper regarding this tool, so we have added the official hyperlink “https://github.com/nanoporetech/minknow_api” as described in previously published papers.

*Xu, W., Zhang, L., Cunningham, A. B., Li, S., Zhuang, H., Wang, Y., & Liu, A. (2020). Blue genome: chromosome-scale genome reveals the evolutionary and molecular basis of indigo biosynthesis in *Strobilanthes cusia*. The Plant Journal, 104(4), 864-879.*

*Perumal, S., Koh, C. S., Jin, L., Buchwaldt, M., Higgins, E. E., Zheng, C., ... & Parkin, I. A. (2020). A high-contiguity *Brassica nigra* genome localizes active centromeres and defines the ancestral *Brassica* genome. Nature plants, 6(8), 929-941.*

Pham, G. M., Hamilton, J. P., Wood, J. C., Burke, J. T., Zhao, H., Vaillancourt, B., ... & Buell, C. R. (2020). Construction of a chromosome-scale long-read reference genome assembly for potato. Gigascience, 9(9), gaaa100.

Lines 541 to Line 543. How were these parameter values optimised? To what extent was the assembly more- or less-accurate and/or contiguous when different parameter values were chosen?

Response: As mentioned in the methodology section, we used the default parameter (read_cutoff = 1k, seed_cutoff = 25211) of Nextdenovo assembler (v2.2, <https://github.com/Nextomics/NextDenovo>). No optimization was performed.

When discovering repetitive elements, how did the authors distinguish between 'repeats' and families of paralogous genes?

Response: The “repeats” and “families of paralogous genes” were identified by using different tools. Tandem repeats were detected across the genome using the software Tandem Repeats Finder (4.07). More details in line 603-611.

While, homologous gene prediction was performed by comparing protein sequences of *Arabidopsis thaliana*, *Liriodendron chinense*, *Persea americana*, *Cinnamomum kanehirae*, and *Oryza sativa* in the UniProt and SwissProt databases (release-2020_05). More details in line 634 to 642.

TopHat2 not Tophat2.

Response: corrected

Pfam not PFAM.

Response: corrected

In the PDF of Figure 1, something has gone slightly wrong with the 'Ma', as if the image has been slightly truncated. Please check whether anything is missing from the figure.

Response: Thanks for pointing this out. The figure has been updated.

Reviewer #3 (Remarks to the Author):

The authors present the first genome of Chloranthales and resolve important outstanding questions on the diversification of angiosperms that were difficult to address prior to the new

data presented. The combination of the Chloranthus genome, robust comparative analyses testing for the contributions of incomplete lineage sorting and hybridization, and patterns of diversification of important gene families make compelling contributions to our understanding of angiosperm diversification. I do not have any substantive changes to suggest.

Response: Thanks for the kind comments.

Reviewers' Comments:

Reviewer #1:

Remarks to the Author:

The authors have carefully revised the manuscript and responded to all questions and comments appropriately. Significant additional insights were added regarding the timing of the LTR evolution and the presence of the P450s with focus on the specialized metabolism. I have no additional comments and am excited to see this excellent work published.

Reviewer #2:

Remarks to the Author:

I wish to thank the authors for addressing and responding to all my comments.

There are no substantial issues outstanding now.

However, I wonder whether there was some misunderstanding about my original question "When discovering repetitive elements, how did the authors distinguish between 'repeats' and families of paralogous genes?" The authors point out that they used two different tools. They used TRF (4.07) for finding tandem repeats. My concern was that some recently duplicated genes might show up in the TRF results and be treated as 'repeats' rather than paralogous genes. This might not be an issue at all, but I invite the authors to consider whether or not it could confound any of their results.

First round revision

When discovering repetitive elements, how did the authors distinguish between 'repeats' and families of paralogous genes?

Response: The “repeats” and “families of paralogous genes” were identified by using different tools. Tandem repeats were detected across the genome using the software Tandem Repeats Finder (4.07). More details in line 592 to 600.

While, homologous gene prediction was performed by comparing protein sequences of *Arabidopsis thaliana*, *Liriodendron chinense*, *Persea americana*, *Cinnamomum kanehirae*, and *Oryza sativa* in the UniProt and SwissProt databases (release-2020_05). More details in line 625 to 634.

Second round revision

Reviewer #2 (Remarks to the Author):

However, I wonder whether there was some misunderstanding about my original question “When discovering repetitive elements, how did the authors distinguish between 'repeats' and families of paralogous genes?” The authors point out that they used two different tools. They used TRF (4.07) for finding tandem repeats. My concern was that some recently duplicated genes might show up in the TRF results and be treated as 'repeats' rather than paralogous genes. This might not be an issue at all, but I invite the authors to consider whether or not it could confound any of their results.

Response: We apologize that we misunderstood this question in the first round of revision. The reviewer’s concern was that tandem repeats may overlap with protein coding genes because some recently duplicated genes of high similarity might be identified by Tandem Repeats Finder (4.07) and treated as 'repeats'. To address this concern, we mapped the all the identified tandem repeats to the total protein coding genes of *C. spicatus*. The results showed that only 0.012% of total TRF length had overlap with protein coding genes (involving 379 coding genes). We also performed the similar blast search in *Arabidopsis thaliana*, *Oryza sativa* and *Liriodendron chinense*, and all these species showed a very low percentage of overlap between tandem repeats and coding genes. Given the very low proportion of genes were annotated as tandem repeats, it is unlikely to confound any of our results at the genome

level. The blast results were incorporated in supplementary table S5. We also add the sentence in the main text for better clarity as ‘The results obtained by Tandem Repeats Finder were mapped to predict coding genes of *C. spicatus* to estimate the proportion of incorrectly detected paralogous genes (Supplementary Table S5).’ Line 134-136

Supplementary Table S5 (partial). Overlap between tandem repeats and protein coding genes in selected species.

	Arabidopsis thaliana	Oryza sativa	Liriodendron chinense	Chloranthus spicatus
Assembly Size (Mb)	119.67	374.47	1742.42	2964.14
Total number of TRF	26,959	138,253	825,718	1,539,137
Total TRF Coverage (bp)	3,305,602	13,024,638	79,438,868	124,686,686
Number of overlapped coding genes	77	185	2250	379
Overlap between coding genes with TRF Regions (bp)	52,000	43,563	85,128	14,995
Overlap ratio in TRF region (%)	1.57	0.33	0.11	0.012

Reviewers' Comments:

Reviewer #2:

Remarks to the Author:

Thank you to the authors for addressing my final remaining concern. I have no further criticisms to make and congratulate the authors on their interesting paper.